



# 1 On the quality of RS41 radiosonde descent data

Bruce Ingleby[1], Martin Motl[2], Graeme Marlton[3], David Edwards[3], Michael Sommer[4], Christoph von
Rohden[4], Holger Vömel[5], Hannu Jauhiainen[6]
[1]European Centre for Medium-Range Weather Forecasts, Reading, RG2 9AX, UK
[2]Czech Hydrometeorological Institute, Prague, 14306, Czechia
[3]Met Office, Exeter EX1 3PB, UK
[4]Deutscher Wetterdienst (DWD)/GCOS Reference Upper Air Network (GRUAN) Lead Center, Lindenberg, Germany
[5]National Center for Atmospheric Research, Boulder CO, 80301, USA
[6] Vaisala Oyj, 01670 Vantaa, Finland
*Correspondence to*: Bruce Ingleby (bruce.ingleby@ecmwf.int)
**Abstract.** Radiosonde descent profiles have been available from tens of stations for several years now - mainly from Vaisala
RS41 radiosondes. They have been compared with the ascent profiles, with ECMWF short-range forecasts and with co-located
radio-occultation retrievals. Over this time our understanding of the data has grown, and the comparison also shed some light
on radiosonde ascent data. It has become clear that the fall rate is very variable and that it is an important factor, with high
fall rates being associated with temperature biases, especially at higher altitudes. Ascent winds are affected by pendulum
motion, on average descent winds are less affected by pendulum motion and are smoother. It is plausible that the true wind
variability in the vertical lies between that shown by ascent and descent profiles. The discrepancy indicates the need for
reference wind measurements.

## 20 1 Introduction

Radiosondes were first developed in the 1930s and have been used to measure profiles of temperature, humidity and wind
since then. There are now approximately 800 operational radiosonde stations worldwide, mostly providing ascents once or
twice per day. These are used for Numerical Weather Prediction (NWP), climate studies and other applications. The Global
Climate Observing System (GCOS) set up the GCOS Reference Upper-Air Network (GRUAN) to produce reference quality
data, with uncertainty estimates, from a subset of stations (Bodeker et al, 2016). Climate users, like GRUAN, tend to focus
on temperature and humidity. For NWP the winds are arguably more important (shown for aircraft data by Ingleby et al, 2021)
- partly because satellites provide more temperature and humidity information than wind information. One attraction of
radiosonde descent data is that there is very little additional cost involved and potentially an extra vertical profile, assuming
that the quality is acceptable. Whilst performing this study, it has become apparent that descent data prompts a re-examination
of ascent data and this can either support or challenge our views of the ascent data.





As radiosondes are designed to measure during the ascent, it is useful to consider how they differ from dropsondes which
always measure on descent. Dropsondes are launched from aircraft and are mainly used for sampling around tropical cyclones
and for field experiments. Radiosondes typically have the temperature and humidity sensors mounted diagonally above the
radiosonde body whereas dropsondes (e.g. Hock and Franklin, 1999) have the sensors underneath - in each case to sample air
undisturbed by the radiosonde body. The AVAPS (Advanced Vertical Atmospheric Profiling System) processing system used
by many dropsondes includes an 'inertial' correction for the delayed response to horizontal wind shear (Appendix of Hock
and Franklin, 1999). Modern radiosondes are usually on a line 30-55 m below the balloon whereas dropsondes are only 1 m
or less below a parachute. As noted by Wang et al (2008) 'The dropsonde fall rate is much smoother than the radiosonde
ascent rate because of the radiosonde's pendulum effect and self-induced balloon motion'. Typically dropsondes fall at about
10 m s$^{-1}$, just after launch it can be about 20 m s$^{-1}$ before the parachute opens fully. As discussed below, radiosonde descent
can be much faster (to 100 m s$^{-1}$ or more if no parachute is used) shortly after balloon burst. There has been some use of
controlled descent, by partial deflation of the balloon, for measurement of stratospheric humidity (Hurst et al, 2011). Zhang
et al (2019) tested the use of a low density 'hard ball' to give more consistent drag than a parachute when deriving the vertical
velocity of the air using a radiosonde descending from a height of about 10 km.
Figure 1 shows BUFR (Binary Universal Form for Representation of meteorological data) descent reports over Europe for
September-November 2019 (descent data were also available from New Zealand, not shown). BUFR allows the reporting of
high vertical resolution radiosonde data (Ingleby et al., 2016). Geller at al., (2021) found that in mid-2020 44% of operational
radiosonde stations were providing high vertical resolution ascent data. Since 2019 descent data has become available from
more European stations and a few in the Americas. After launch the balloon is advected horizontally by the wind, especially
at upper levels, and typically travels 50 to 300 km before burst with the larger distances being more common in winter (Seidel
et al., 2011).





## Sep-Nov 2019: Descent data BUFR availability/type

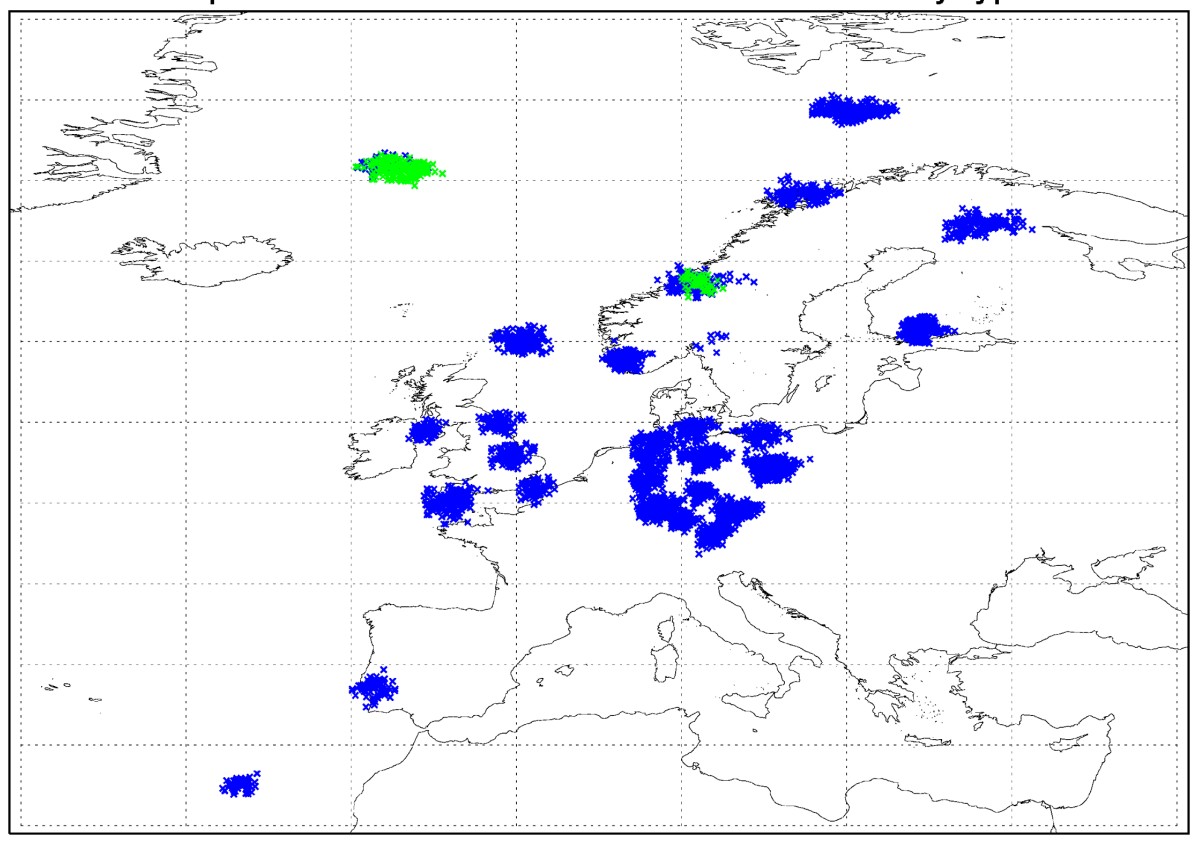

**Figure 1: Descent reports (burst positions) over Europe for September-November 2019, blue - Vaisala RS41, green - Modem M10. There were 14 stations from Germany, 6 each from UK and Norway and 2 each from Finland and Portugal.**

Figure 2 gives an indication of the number and vertical extent of descent profiles. Larger balloon size and fill volume is used to achieve higher altitudes. On average, radiosondes that achieve higher altitudes drift further horizontally, resulting in the radio signal to the launch station being lost at higher altitudes on descent due to obstruction by terrain or signal attenuation. This can be seen clearly in the UK results which have been split into automatic and manual launches: the manual launches use larger balloons and the number of descent reports starts to decline earlier, below 9 km. Automatic launchers are documented by Madonna et al. (2020). Some of the other countries use a mixture of manual and automatic launchers, but with little or no difference in balloon size. A small proportion of ascents do not have a corresponding descent report, often due to a fault developing with the radiosonde before or upon burst, leading to an automatic termination.

.

| Country | Parachute | Pressure sensor | Balloon weight (g) |
|---|---|---|---|
| Norway | No | Yes | 350 (01004*: 1500) |
| Finland | No | No | 350 |
| UK | Yes | No | 350 (03005, 03808: 800+) |





| Germany | Yes | Yes | 600 (also 300, 800) (10962: 1200+) |
| Portugal | Yes (2 stations) | Yes | 600 |
| New Zealand | No | No | 350 |

Table 1. Summary of metadata for countries providing descent data in 2019. For balloon weight the most common value is given followed by others used in brackets, usually with an indication of the stations involved. (* station 01004 did not provide descent reports in 2019.)

Figure 2: Numbers of RS41 descent reports by height and country, September-November 2019.





70

## 2 Radiosonde ascent and descent

### 2.1 What goes up must come down

A balloon is filled with hydrogen or helium and ascends, attached by a string to the radiosonde (instrument package). Balloon techniques are documented by WMO (2018b). The Vaisala RS41-SG radiosondes have a small sensor boom with temperature and humidity sensors near the end, and wind and position are measured using a GPS receiver. Some models have a pressure sensor, identified as the RS41-SGP (pressure is discussed in section 4.2). The measurements are transmitted back to the ground station and processed there. Dirksen et al (2014) describe the GRUAN processing of the Vaisala RS92 and the instrument accuracy; the operational BUFR reports come from the Vaisala processing. The Vaisala RS41 is the successor to the RS92 and is similar in many respects, but with improved humidity and temperature measurements (Edwards et al, 2014; Jensen et al, 2016). As the balloon ascends it expands in diameter and eventually bursts causing the radiosonde to descend - transmission of the measurements continues but traditionally processing stops at this point. When the radiosonde falls below the horizon as seen from the ground station then it is no longer possible to receive the transmissions. Typically, the ascent takes 90-120 minutes (reaching altitudes of 30 or 35 km) and the descent takes 30 minutes or less. The upper part of the descent is close to the upper part of the ascent in both time and space, usually with increasing separation as the radiosonde descends.

Some operators include a parachute, either inside or just below the balloon. The parachute slows the descent and is intended to reduce the risk of damage to life and property when the radiosonde reaches the surface. In sparsely populated or island countries a parachute may not be used.

From rare images of the balloon burst (figure 3) and recovered radiosondes (mainly those launched from Lindenberg, Germany with extra instruments), shown in figure 4, and also from the motion on descent it is clear that a) sometimes the balloon bursts completely or tears off at the nozzle and the parachute opens fully, b) sometimes the balloon tears open leaving strips attached, these may get tangled with the parachute - which may partially 'free itself' later, c) sometimes the parachute ruptures and so is ineffective. Where there is no parachute we speculate that sometimes the remains of the balloon act to slow the descent. Note also that when complete, the mass of the balloon is typically several times that of the radiosonde (larger balloons are used to reach higher altitudes, they are also sometimes used at night; the same balloon/amount of gas will reach higher in the daytime on average). Some stations add extra instruments periodically, for example once a week Lerwick (03005) measures ozone as well and a larger balloon and parachute are used.




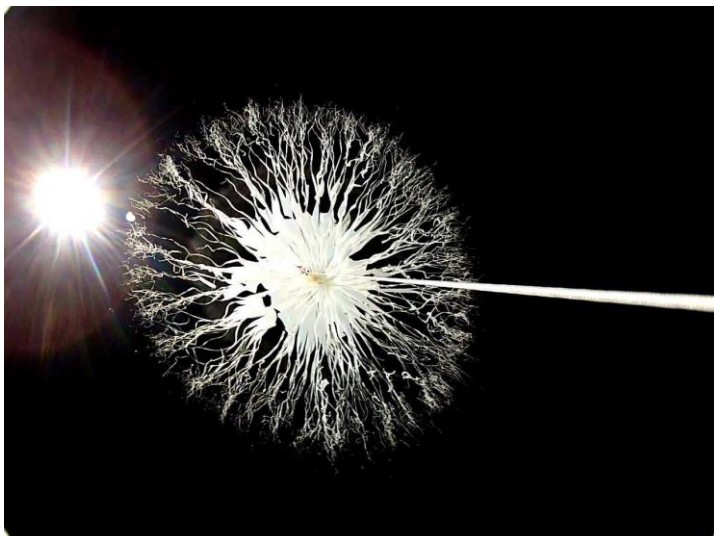


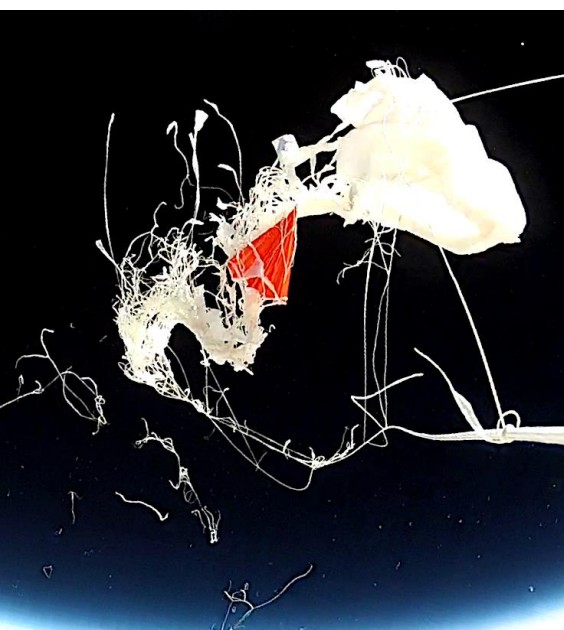


Figure 3. Photographs of a bursting balloon (top) and parachute (orange) entangled in balloon remains (bottom). Balloon and
parachute were used for a frostpoint hygrometer launched at Lindenberg and are larger than those for a regular radiosonde
launch.



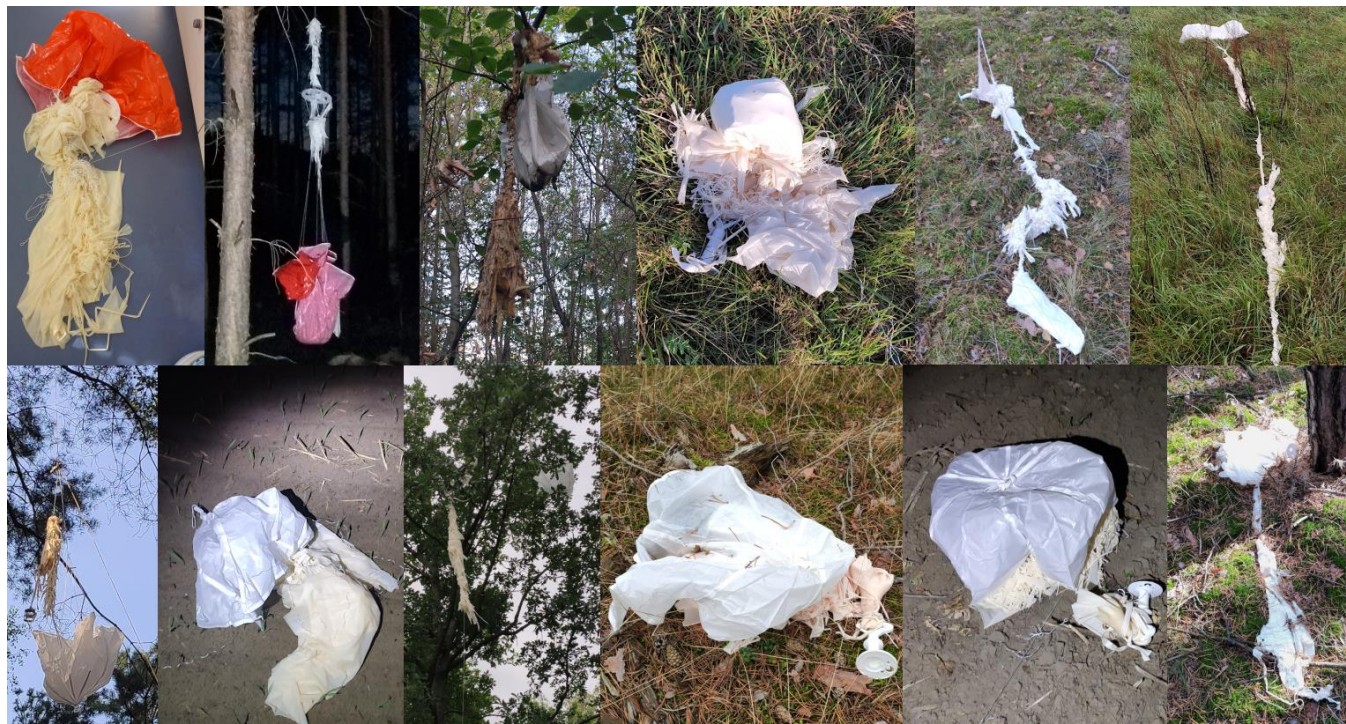

Figure 4. Collage of photographs showing balloon remnants (Totex TA and TX balloons of different size) and parachutes (orange or silvery) after landing - Lindenberg launches.

On ascent the sensor boom projects above the radiosonde, so that it samples air that has not flowed over the body of the radiosonde. On descent, with a working parachute, it should be in a similar position - so it may sample air that has flowed over the radiosonde body, which has the potential to introduce biases or contamination. It is not known how a radiosonde descending without a parachute is orientated, or if it may be tumbling.

**2.2 Types of parachute and string length**

For some manual launches a parachute (if used) is attached to the line not far below the balloon. Figure 4 shows that sometimes the two can become entangled after balloon burst. For automated launches a parachute (if used) is stored within the balloon and this seems to cause fewer entanglement problems. This can be used for manual launches too, and has been used at Lindenberg for some years, but there is a small additional expense. In general, most of the parachutes are quite basic and do not include a hole. Air can build up inside the parachute and suddenly spill out. It is clear from some of our results that parachutes do not always open as intended.

In earlier decades the string connecting the balloon and the radiosonde may have been 10 m or less, but in the stratosphere the balloon gets larger and there can be intermittent influences of the balloon wake upon the instruments (WMO, 1994; Luers and Eskridge 1998; Söder et al 2019). For this reason longer suspensions are used now, WMO (2018a) suggests 40 m for





radiosondes ascending to 30 km or higher. In practice an 'unwinder' is used to increase the line length shortly after the
radiosonde launch (WMO, 2018b). The Vaisala unwinder for the RS41 gives 55 m when extended (Vaisala, 2017). We note
that while longer lines benefit stratospheric temperature measurements they cause larger amplitude pendulum motion in the
winds.

### 2.3 Preparation of profile reports

Data values are transmitted to the ground receiver every second and are processed by Vaisala MW41 software. Raw data
values, both ascent and descent, indexed by time are stored locally (the GRUAN archive makes the one second data available
for GRUAN sites). The MW41 software looks for a sustained decrease in altitude to determine the time of burst. In the past,
all later data would usually have been discarded but there is now an option to continue processing and to produce a separate
BUFR descent message using sequence 3 09 056 (WMO, 2019) designed for descent data. In 2019, as an interim measure, the
dropsonde sequence 3 09 053 was being used. As the timeliness of ascent data is critical for data users, it is preferable to
transmit the ascent profile as soon as possible after burst, followed by the descent data sent once the profile is completed.
BUFR from the European stations involved in this study is generally provided every two seconds (about 10 m separation in
the vertical during ascent).
The MW41 horizontal winds are derived primarily from Doppler processing of the Global Positioning System (GPS) signals
but the GPS locations are also used (GRUAN processing only uses the GPS positions). There is very good vertical resolution
but it also means that the winds sample the pendulum motion of the radiosonde - this is probably a mixture of planar and
conical pendulum motion. The period of the pendulum motion is a function of the length of line between the balloon or
parachute and the radiosonde. The processing attempts to filter out the pendulum motion (discussed briefly in Dirksen et al
2014), but the filtering is imperfect as discussed below.

### 2.4 Descent fall rates

The balloon and gas are chosen so that the ascent rate is about 5 m/s on average - however there is usually notable high
frequency variability probably due to pendulum motion. Especially in the stratosphere, there can be lower frequency signals
due to gravity waves and both of these features can be seen in Figure 5 (grey line, ascent). After the balloon bursts the
radiosonde falls very fast (over 70 m/s in this case) and then slows abruptly - presumably when the parachute opens fully.
After this there is a little high frequency variability (but much less than on the ascent) and a gradual decrease in descent rate
as the air density increases. Looking at a sample of Lindenberg descents over several weeks (supplementary material), some
exhibit an abrupt deceleration and others do not. Figure 6 shows descent rates from the station at Sola in Norway, without
parachutes. These do not show the abrupt deceleration, but do show a large range of descent rates. The slower descents tend
to have larger high frequency variability. We tentatively suggest that in these cases, the remains of the balloon are acting to
slow the descent and there is some pendulum motion. The variability in the descent rate may be due to variability in the mass



and shape of balloon remnants. Similar figures for several weeks of descents at Lindenberg are shown in the supplemental
material.
Mean descent rates by country are shown in Figure 7 (it should be remembered that this hides a lot of variability). For any
particular altitude those from Germany and the UK are slowest, reflecting their use of parachutes. Amongst the others there
is a large range. The Norwegian radiosondes fall faster than those from the other countries studied - it is unclear why they fall
faster than the Finnish radiosondes. In section 3 we focus on the four northern European countries (Norway, Finland, UK and
Germany) because they have similar upper air climatologies but different instrument characteristics. For Germany and Finland
the descent data received at ECMWF stopped on about 20 November 2019 linked to the move to the new BUFR template.
During 2020 the volume of descent profiles increased overall (e.g. France and Spain started sending them), although there was
also some disruption from the Covid pandemic.





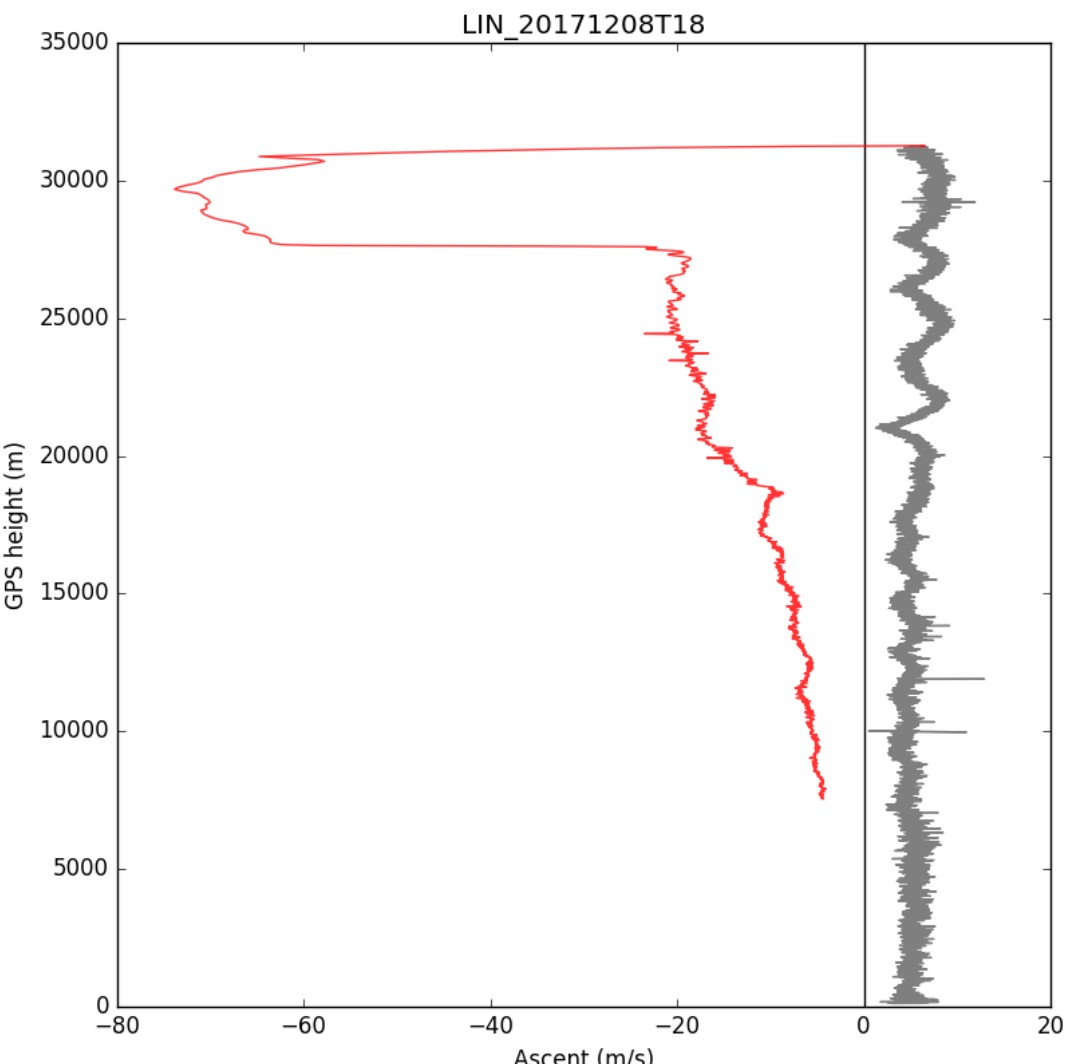

**Figure 5: Ascent (grey) and descent (red) speed - an example from Lindenberg (1 s data).**

**Figure 6: Descent profiles from Sola (Norway, data courtesy of Terje Borge): 14 December 2019 - 5 January 2020.**



Figure 7: Mean descent rates for September-November 2019, same categories as Figure 2.



## 2.5 Motion of radiosonde during descent

A radiosonde as it ascends through the atmosphere can be thought of as a pendulum with a moving pivot (Marlton et al, 2015). As the radiosonde encounters small scale turbulence which is ubiquitous in our atmosphere it causes the radiosonde beneath to swing. The periodicity $\tau$ is a function of string length $l$ given by

$$\tau = 2\pi \sqrt{\frac{l}{g}}, \quad (1)$$

where $g$ is the acceleration due to gravity. Different radiosonde manufacturers supply different string lengths to their radiosondes, with the aim of removing the radiosondes sensors from the wake effects (Luers & Eskridge 1998). The standard string length on the Vaisala RS41 is 55 m (Vaisala 2017) which gives an approximate period of oscillation 14.9 seconds and an oscillating frequency of approximately 0.06 Hz. Differing balloon sizes and the inclusion of a parachute may alter $l$ and therefore $\tau$ slightly, a +/- 5m variation of $l$ affects $\tau$ by +/- 0.6 s. Depending on the operating practices the radiosonde may be launched in three broad configurations: i) No parachute; the radiosonde freefalls with some drag from the balloon remnants, ii) Balloon bursts above the parachute and radiosonde descends on the parachute and iii) The balloon contains a parachute which then deploys above the neck of the balloon and similarly descends. In addition to this, the deployment of the parachute is not consistent, see Figure 5 and Figures S3 and S4 in the supplemental material.

Marlton (2016) performed a spectral analysis of raw GPS wind measurements from Vaisala RS92 radiosonde ascents equipped with motion sensors described in Harrison and Hogan (2005) and Marlton et al (2015) and found oscillatory modes detected by the motion sensors were present in the raw GPS data. In this section raw GPS ascent and descent data from UK Met Office Autosonde sites and manned stations are used to generate Lomb Periodograms of the raw horizontal wind components.

Due to radiosondes often travelling 4-5 times their vertical ascent height in the horizontal there are on occasion small data gaps due to transmission drop out. The issue becomes more noticeable in descent data as the radiosonde is now even further from its ground station. This means a traditional Fourier transform method is not appropriate. Thus, a Lomb periodogram is chosen (Lomb 1979), which can generate periodograms which have irregularly sampled data. To ensure that we focus on the motion of the radiosonde we use the processed horizontal wind components to remove the wind field from our raw GPS readings leaving the motion of the radiosonde beneath that balloon.



Figure 8. Composite Lomb Periodograms of detrended horizontal GPS data as a function of height for ascent data from RS41's
with the following launch configurations a) TX350 balloon with internal parachute, b) TX500 with no parachute, c) TX1200
with external parachute (day time only) and d) TX800 with external parachute. Panels e-h show composite Lomb periodograms
of descent data from the balloon configurations a-d respectively. Profile contributions for balloon configurations a-d during
ascent (descent) are shown in black (red) in panels i-l respectively.



Figure 8 a-d shows Lomb periodograms of the detrended horizontal GPS during an ascent for four different RS41 launch
configurations, in each case there is a dominant oscillatory period of 15 s (0.06 Hz) which strongly dominates above 15 km.
Examining the results from Eq (1) given the RS41's string length shows that on ascent the radiosonde and balloon are behaving
as a pendulum with a moving pivot.

During descent the oscillatory motion is very different, there is no longer a dominant oscillatory period and the amplitudes of
these oscillations are smaller. A general trend is that in the early stages of the descent the radiosonde is still oscillating with a
period of 15 s (0.06 Hz). As it falls the peak period of oscillation increases to 25-30 s, until the height of 15-20 km. At this
approximate height the non parachuted RS41s (panel (e)) exhibit a narrow spectral width with the smallest descent to descent
variability. An oscillation is still present indicating that some of the balloon remnants are acting as a parachute. For the
parachuted RS41s the spectral width in oscillation widens significantly indicating that there is variation in the motion behavior
of the radiosonde. As discussed earlier this may be due to how and when the parachute deployed and if any of the parachute
remains entangled with the parachute rigging. The latter is hard to determine without retrieving the radiosonde which is seldom
done. We can get a better understanding of the variation in oscillation by looking at individual ascents.

Figure 9 and Figure 10 show two descents from Castor Bay Autosonde station (54.50 N, 6.34 W). In both figures panel a
shows the processed horizontal wind components $u$ and $v$ components in red and blue respectively. The raw GPS wind
components are shown in black and orange for the $u$ and $v$ components respectively. Panel b shows the descent speed and c
and d show Lomb periodograms of the detrended raw GPS velocities. In Figure 9 we can see that the parachute does not seem
to offer significant deceleration to the sonde until about 14 km. During the descent there are weak low frequency oscillations
greater than 60 seconds over the duration of the ascent. In panel figure 9a the raw GPS and the processed $v$ component of the
wind track very closely and it is hard to differentiate between them.  Figure 10 is a descent from a different day which tells a
very different story. The parachute deploys within 1 km of the burst height and causes a sudden deceleration from -60 m s$^{-1}$ to
-20 m s$^{-1}$. After the rapid deceleration the radiosonde enters a high amplitude oscillatory mode with a periodicity of 30-40
seconds as it descends. A hypothesis here is that the sudden deceleration caused by a correct deployment of the parachute has
caused the oscillatory mode seen here. The amplitude of oscillations seen under this scenario could introduce error in the
processed winds and is a possible area for future study.

Here it has been shown that identical balloon configurations have very different and random descent characteristics. Figure 4
shows that the balloon sometimes intertwines itself about the parachute which may affect how well the parachute deploys and
in turn its oscillatory characteristics on descent. A successful parachute deployment can enhance the oscillation such that it
has potential to introduce error in descent wind data. More research in this area needs undertaking using an approach where
motion sensors are attached to the RS41 to better understand the descent to descent variability. Additional investigations where
guillotines cut the balloon from the parachute such as that used on heavy scientific balloon payloads, could be utilised to



remove the effect of balloon entanglement. The placing of a small central hole in the top of the parachute to improve stability
and removal of sudden deceleration also need investigating.


Figure 9.  Vertical profiles of a) processed horizontal wind components *u* and *v* in solid red and blue  respectively with raw
GPS winds in black and orange for the *u* and *v* components respectively, and b) descent speed. Panels c) and d) show Lomb
periodograms of the detrended raw GPS velocities as a function of height for a sounding made at Castor Bay autosonde station
(54.50 N, 6.34 W) at 2315UT on 23/12/2020.



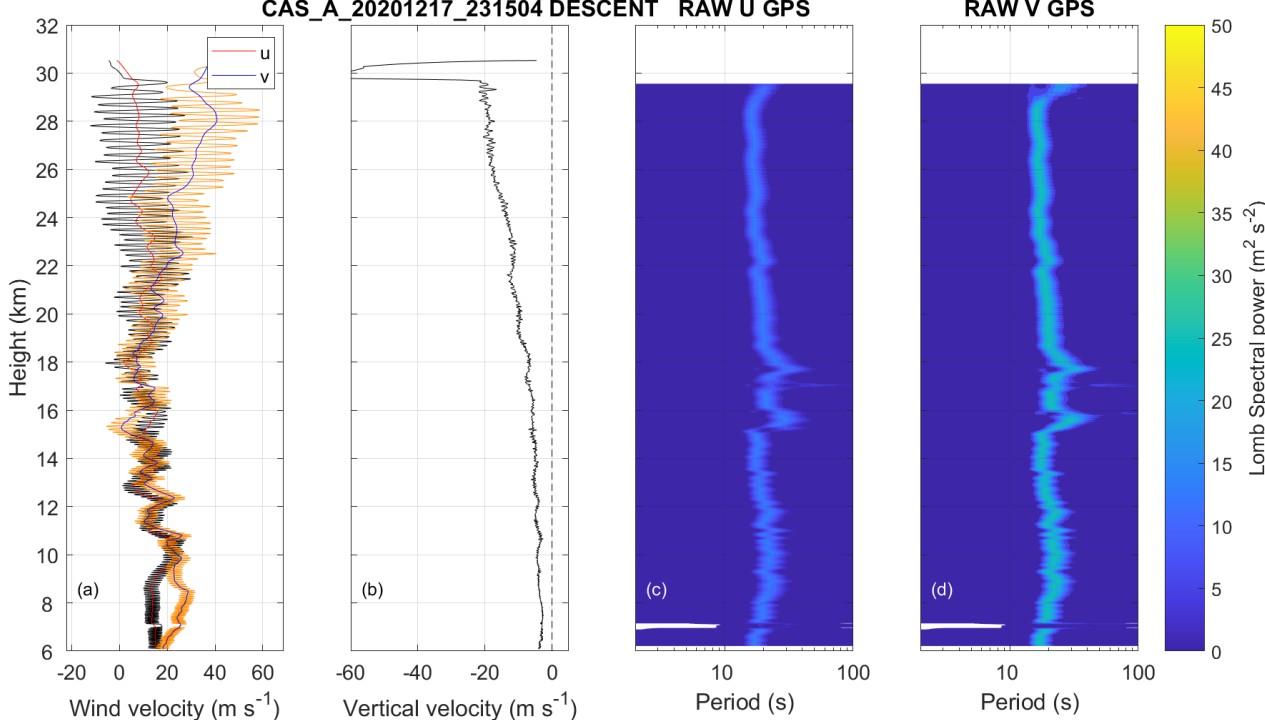

Figure 10. Vertical profiles of a) processed horizontal wind components in solid red and blue for u and v respectively with raw GPS winds in black and orange for u and v respectively, and b) descent speed. Panels c) and d) show Lomb periodograms of the detrended raw GPS velocities as a function of height for a sounding made at Castor Bay Autosonde station (54.50 N, 6.34 W) at 2315 UTC on 17/12/2020.

In summary, ascending radiosondes tend to have similar characteristics in terms of motion beneath the balloon and ascent speeds, although the latter does depend on the amount of gas within the balloon. Descending radiosondes have widely varying descent characteristics which are due to the random nature of how the balloon and parachute interact (if present) and how effective the parachute is at slowing the balloon. The motion on descent may be more consistent if the radiosonde could be 'cut free' of the balloon remains and fall on its own without a parachute. It would be interesting to study the effect of cutting the string just after balloon burst, but this may be technically difficult and the risk associated with the radiosonde falling at terminal velocity would need to be assessed. Given the variation in burst heights, reliably cutting it before burst would reduce the average height attained. In addition similar motion and orientation sensors as used in Harrison & Hogan (2004) and Marlton et al. 2015 could be used to ascertain more information about the orientation of the descending radiosonde package



## 3 Comparison with ECMWF background fields

### 3.1 ECMWF forecasting system

For comparison we use statistics from the ECMWF operational data assimilation system for September to November 2019. The forecast model had a horizontal grid spacing of about 9 km and 137 levels in the vertical and the assimilation used 4DVar with a 12 h window. The 3...15 h forecast from the previous analysis forms the background for the assimilation and the observation-minus-background (O-B) statistics can yield a lot of information. The background values are not perfect but provide a relatively accurate and (generally) independent estimate of the measured variables. In many respects the forecasting system is similar to that of ERA5 (Hersbach et al, 2020) which was based on the operational system of 2016. One difference from ERA5 is that treatment of radiosonde drift was introduced operationally in June 2018 and this improved upper-level O-B standard deviations by 5-10% (Ingleby et al, 2018). Prior to this radiosonde profiles were treated as vertical and instantaneous, afterwards the profiles were split into sub-profiles of 15-minutes each and treated as valid at time and latitude/longitude of the first point in the sub-profile. Descent profiles are split into 5-minute sub-profiles for comparison with the model.

### 3.2 Wind comparison

Figure 11 shows mean and standard deviation (SD) profiles of O-B differences at radiosonde standard levels for the u (zonal) component of the wind. The statistics for the v (meridional) component are similar and are not shown. The mean differences (dashed lines) are close to zero, as hoped. The standard deviations are approximately 2 m s$^{-1}$, but are slightly larger at the top levels. One surprise was that the descent profiles (in red) fit the background more closely than the ascent profiles (in black), particularly at upper levels. Comparing individual ascent/descent profiles the descent winds generally appear smoother and this appears to be the cause of the better fit to background. Figure 12 shows the raw 1-second data for a single profile (faint line) and the data after smoothing to remove the pendulum motion (bold line). In this case the smoothing was performed using the GRUAN algorithm (Dirksen et al, 2014), whereas the BUFR reports have smoothing applied by Vaisala MW41 software which is similar but not identical. In both cases the smoothing is a time filter applied to ascent and descent data in the same way. The period of the pendulum motion depends on the length of the pendulum which will be approximately the same for a parachute as for the balloon. (Of course, if there is no parachute there is a different scenario for the descent.) Because the radiosonde is falling faster than it ascended, a filter based on a fixed time interval corresponds to a larger height interval on the descent. Note also that the MW41 processing does not include an inertial correction as used in the AVAPS dropsonde processing (Sect. 1), this counteracts time-lag effects which will be largest when falling fastest. As shown in Figure 12, at most levels there is less high frequency 'pendulum' motion on the descent - although at the top levels there can be substantial amounts of noise.

Figure 13 shows SD(O-B) for individual descents in the interval 30-50 hPa against the mean descent rate for this pressure range. The standard deviations are slightly larger for slower descent rates, this is thought to be linked to more pendulum motion





when the parachute is slowing the descent more effectively.  Similar effects can be seen for other pressure ranges, but there is
no clear dependence of the mean (O-B) winds on descent rate (not shown).





Figure 11. U-component standard level statistics of mean (dashed) and SD (solid) O-B differences for ascent (black) and
descent (red) for four different countries, September - November 2019.

Figure 12.  Raw (1-second) data (pale line) and filtered (bold line) u component as a function of time: ascent (black) and
descent (red).  This is for a launch from Lindenberg, including a parachute for the descent.











Figure 13 Standard deviation of (O-B) plotted against mean descent rate, both for descents from 30 to 50 hPa (blue symbols).
The green symbols show average values for bins of 10 m s-1.

### 3.3 Temperature comparison

Firstly, we note that at about 50 hPa, in the extratropics, the ECMWF background is too cool by about 0.5°C (this can be seen
against the RS41 ascent data in Figure 14).  This is recognised as a model error, due mainly to excessive humidity and hence
extra long-wave cooling as shown by Shepherd et al (2018).  More recent work on the analysis system has approximately
halved the short-range forecast bias (Laloyaux et al., 2020).  To provide reassurance ascent/descent pairs were compared to
radio occultation (RO) retrievals (Table 2). The RO data (Laloyaux et al., 2020) is much closer to the ascent temperatures than
the descent temperatures - note that the sample size is much smaller than for the O-B statistics.

| Pressure (hPa) | Sample | Ascent-RO (ºC) | Descent-RO (ºC) | Ascent-B (ºC) | Descent-B (ºC) |
|---|---|---|---|---|---|
| 5 | 22 | -0.07 | 1.07 | -0.37 | 0.90 |
| 10 | 36 | 0.53 | 1.63 | 0.25 | 1.25 |
| 20 | 125 | 0.13 | 1.04 | 0.37 | 1.33 |
| 30 | 130 | 0.15 | 0.92 | 0.45 | 1.24 |
| 50 | 135 | 0.02 | 0.37 | 0.44 | 0.84 |
| 70 | 137 | -0.11 | 0.17 | 0.39 | 0.68 |
| 100 | 136 | 0.28 | 0.41 | 0.31 | 0.51 |

Table 2. Collocations with radio occultation retrievals (within 100 km and 2 hours) at standard levels, with mean temperature
differences (°C).  Columns show Radiosonde Ascent (or Descent) minus RO or Background values.
The clearest difference between ascent and descent is that at upper levels the descent temperatures are higher than the ascent
values (Figure 14). This has been noted previously, for different radiosonde types, see section 6. At 10 hPa the descent-ascent
difference is over 1.5°C for the Norwegian stations and about half that for the UK and German stations.  For Finland the
highest standard level reached is generally 20 hPa and the difference there is about 1°C.  One hypothesis advanced was that
this could be a time-lag effect. However, descending from 30 to 100 hPa the mean temperatures were approximately constant
or increasing slightly (Figure 15) suggesting that another explanation is needed.  A convincing link to the radiosonde fall speed
was found (Figure 16).  There is no clear link to the time of day (and solar radiation) as shown by the different coloured
symbols in Figure 16.  The fall rate correction put forward in the next section (derived originally for a single radiosonde station)
does a good job of removing most of the bias (Figure 17).  The SD(O-B) for temperature shows no clear link to fall rate (not
shown).
Returning to Figure 14 the large top-level ascent-descent difference in the Norwegian data has disappeared by 300 hPa, but
the smaller top-level Finnish difference becomes an offset of 0.2 or 0.3°C throughout the troposphere.  The important
difference seems to be that the Norwegian radiosondes have a pressure sensor, but the Finnish radiosondes do not.  Without a





pressure sensor the pressures must be computed and biases in the temperature will feed into later biases in the pressures -
discussed in more detail in the next section. A smaller version of the same effect can be seen between the German data (with
pressure sensors) and the UK data - without a pressure sensor these have an offset of about 0.1°C in the troposphere: smaller
than the Finnish data because the UK radiosondes have parachutes.





Figure 14. As figure 11 but for temperature.



**Figure 15.** Mean ascent and descent temperatures at standard levels - range chosen to emphasise upper levels. Particularly above 30 hPa the locations sampled will be from a smaller subset of profiles/stations.






Figure 16. Comparison between mean fall speed from 20 to 30 hPa and mean O-B temperature. Red markers denote nominal 12 UTC profiles, dark blue markers nominal 00 UTC profiles and cyan denotes intermediate profiles. (Recall that the B values have a bias of about 0.4°C at these levels.)    Green markers show values averaged over all times of day in bins of 10 m s-1.




Figure 17. As Figure 16 but temperatures adjusted as described in section 4.1 (with A=0.0004)..





### 3.4 Humidity comparisons


Figures 18 and 19 show ascent/descent comparisons with the background for specific and relative humidity (RH). Broadly
speaking the ascent and descent statistics are very similar, although the descent fit to background is slightly worse for the
Finnish radiosondes in the troposphere. Between about 50 and 150 hPa the SD(O-B) for RH is smaller for the descent, but
note that stratospheric radiosonde humidity is not assimilated in the ECMWF or other NWP systems.






Figure 18. As figure 11 but for specific humidity.

Figure 19. As figure 11 but for relative humidity.





## 4 Warm bias during descent

### 4.1 Direct effect of heating

As the comparison of descent data with NWP model suggests, there is a positive temperature bias for the data measured by descending Vaisala RS41 radiosondes. This bias is bigger in the stratosphere than in the troposphere and is more significant for the data taken from radiosondes without parachutes.

As the descent rate often exceeds 50 m s$^{-1}$, occasionally even 100 m s$^{-1}$, frictional heating seems to be a reasonable explanation of the observed bias. A related issue is recognized for sensors on aircraft, which also measure temperature while moving with high speed relative to the free air (WMO 2018b, section 3.3). For aircraft the kinetic energy is transferred to internal heat mostly by adiabatic compression. In the case of radiosondes we expect that most of the conversion is done by direct collisions of air and sensor molecules (friction), but it is also possible that the effect is done by adiabatic compression in the boundary layer of the sensor. We use a quadratic relationship on descent rate (DR) - this arises from a simple energy balance, independent of the energy conversion mechanism:

$$\Delta T = A.DR^2 \quad (2)$$

A is a coefficient, determined below. This is similar to the equation for the heating of aircraft temperature sensors - see Appendix. It is also linked to the 'viscous dissipation' or 'compressional heating' mentioned by Wagner (1964) for rocketsondes (launched high in the atmosphere by a rocket they measure on the descent, slowed by a parachute). This relationship was examined by comparing the descent temperatures with ascent temperatures from the same radiosonde. Most of the data were from the Praha-Libus (Prague) upper air station: 554 descents with average length of descent 23 km. From these data there was a sample of about 528 000 comparable levels. The data covers the period from July 2019 to January 2020. This station used radiosondes RS41-SG, without pressure sensor or parachute - so similar to the soundings from Finland in section 3.

The time and space difference between ascent and descent measurement of a particular level starts at 0 km and seconds at the moment of balloon burst and can rise up to 2 hours and 150 km for lower troposphere levels. We expect that this difference will result in deviations of atmospheric measurements, but according to the long-term data there wasn't expected any bias caused by this difference in the stratosphere. For the lowermost layers (below 4 km) was expected warm bias for 06 and 12 terms, and cold bias for 00 term due to the diurnal variation.

For each level of descent was taken height (H), descent rate (DR) and descent temperature ($T_D$), with ascent temperature ($T_A$) interpolated to this level. After dividing the sample into groups of 1000 m the bias was calculated (mean temperature difference $\Delta T = T_D - T_A$) for each of these groups. Results shown in Figure 20 are very similar to comparison of German data shown in Figure 14 – about 1°C bias at the highest levels decreasing to 0°C at 12 km. A positive bias near the surface is an expected effect of diurnal variation and after separating data into 00, 06 and 12 UTC groups, the differences seen are approximately -0.4°C, 0.2°C and 1.1°C respectively.



According to equation [2], ΔT should depend solely on DR. Pearson's correlation coefficients confirms the strong link between
those two variables. It was 0.21 between ΔT and H, between ΔT and DR it was 0.40.

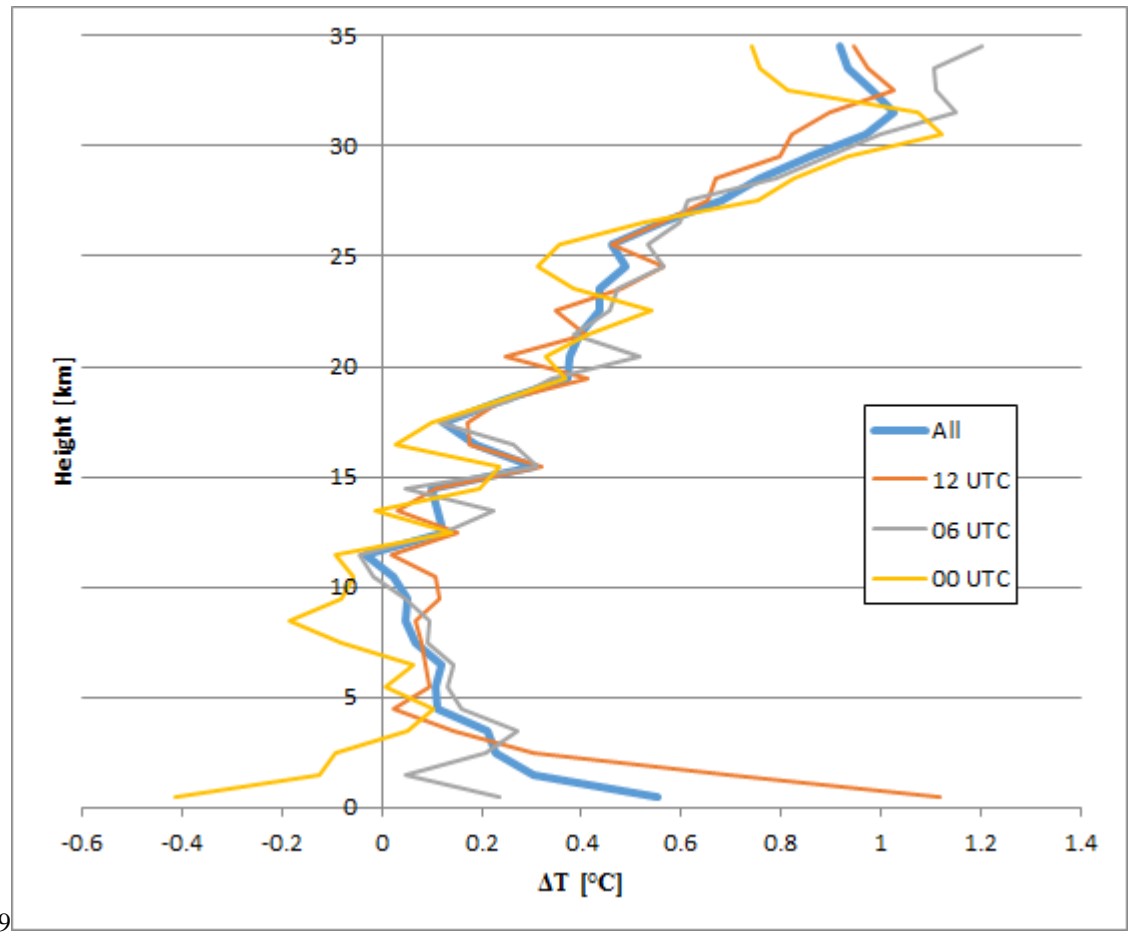

394 9

Figure 20. Temperature differences between ascent and descent as a function of height (Praha-Libus data).

In the next step the sample was binned by DR – intervals used were 0-5 m/s, 5-10 m/s etc. There is clearly a quadratic
dependence of ΔT on DR in Figure 21 (average ΔT for these bins). The standard deviation of ΔT shown with grey lines is
almost independent of DR. The black line is the best estimate with A= $4.05 \cdot 10^{-4}$.
For DR greater than 110 m/s the fit is slightly less good but the sample size is small with data available from less than 3 % of
examined soundings. When equation [2] with A= $4.05 \cdot 10^{-4}$ is applied as a temperature correction, the root mean square ΔT
is lowered from 1.22 °C to 1.06 °C, indicating that the correction explains 24.4 % of the variance seen.
Calculating the correction as a complete quadratic equation (ΔT = $4.39 \cdot 10^{-4} \cdot DR^2 - 3.17 \cdot 10^{-3} \cdot DR + 5.40 \cdot 10^{-2}$), did not
significantly improve the result (explained variance increased by less than 0.01 %).
To find out if the result was affected by lower tropospheric differences (which are mostly caused by diurnal variation and not
friction), the result was recalculated for the sample with all data below 4 km excluded. Results changed only very slightly
again, the coefficient was then A = $4.04 \cdot 10^{-4}$·, and the explained variance increased to 25.3 %.

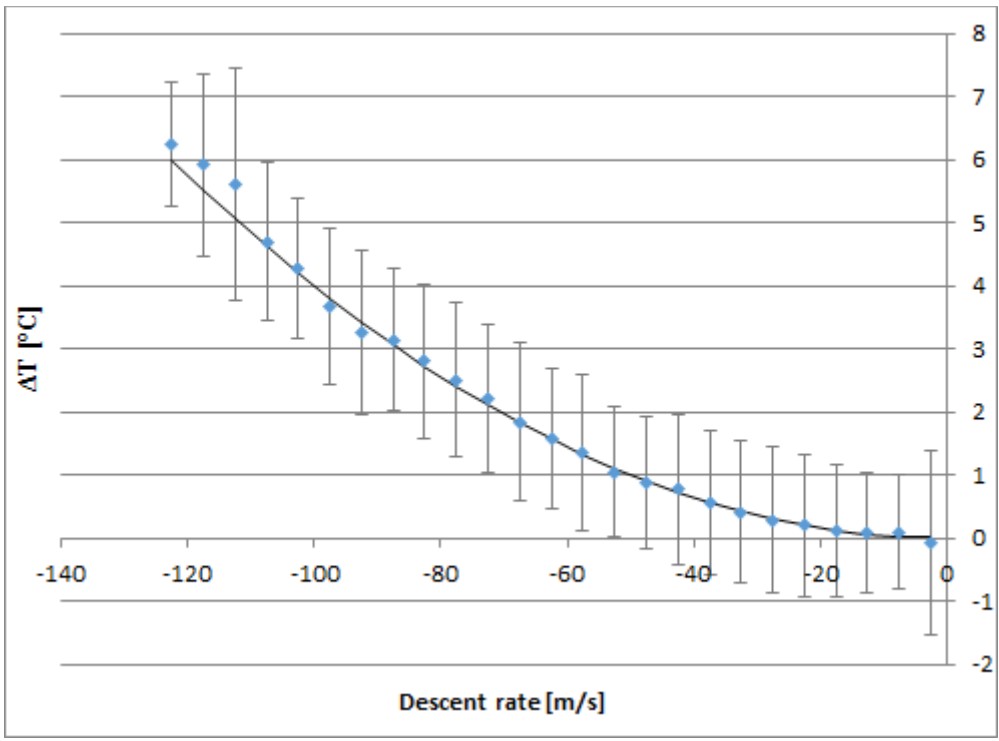

Figure 21.  Dependence of temperature differences between ascent and descent on descent rate, Praha-Libus.

The coefficients were also calculated separately for the data from each time of the launch
— 00, 06 and 12 UTC soundings — the estimates of coefficient A range from $3.9 \cdot 10^{-4}$ to $4.3 \cdot 10^{-4}$
(Table 3).

| Best estimate (at time, UTC) $\Delta T = A \cdot DR^2$ | A [$°C \cdot s^2 \cdot m^{-2}$] |
|---|---|
| 00 | $3.90 \cdot 10^{-4}$ |
| 06 | $4.22 \cdot 10^{-4}$ |
| 12 | $4.07 \cdot 10^{-4}$ |

Table 3. Best estimate of correction coefficient for different times of launch at Praha-Libus.






Figure 22 shows the mean and SD of ΔT as a function of height before and after applying the correction. We can see that bias
was almost completely removed, except for the lowest layers, where the bias is expected due to diurnal effects. Another notable
result was that ΔT SD for heights above 20 km was significantly lowered.

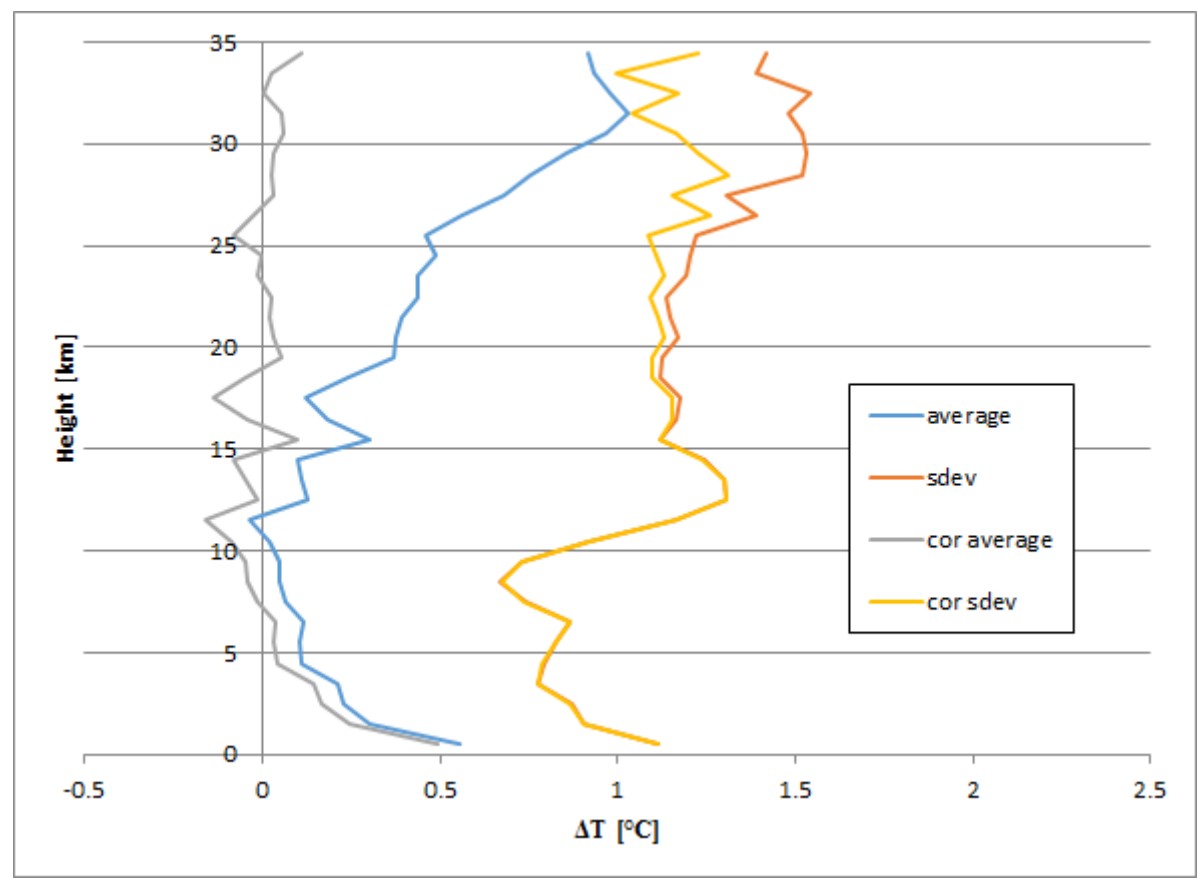


Figure 22. Average ΔT and its standard deviation before and after friction correction (Praha-Libus).

Investigations were extended to other stations to examine consistency. Information about type of the data, data sample and
calculated coefficient are in Table 4.

| Station | Praha-Libus | Lindenberg | Sola |
|---|---|---|---|
| Country | Czechia | Germany | Norway |
| Radiosonde | RS41-SG | RS41-SGP | RS41-SGP |
| Parachute | no | yes | no |
| Sample start | Jul 2019 | Nov 2019 | Dec 2019 |



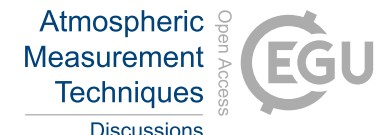

| Sample end | Jan 2020 | Feb 2020 | Jan 2020 |
|---|---|---|---|
| Soundings | 554 | 329 | 45 |
| Compared levels | 527 779 | 650 399 | 37 670 |
| Coefficient A | $4.05 \cdot 10^{-4}$ | $4.46 \cdot 10^{-4}$ | $3.44 \cdot 10^{-4}$ |

Table 4: Coefficient A determined from different samples

It can be seen from the results in Table 4 that the exact value of the correction coefficient is slightly uncertain. As there is a lot
of noise in the data due to other reasons of ΔT than just friction, we would need a larger sample of data to investigate further.

**4.2 Indirect effect of heating**
Some radiosondes measure atmospheric pressure using a sensor and the geopotential height is calculated using hydrostatic
equilibrium: $dP = -\rho(H) g\, dH$, where density of air $\rho$ depends on pressure, temperature and humidity of the air. This method is
used for processing of the data from RS41-SGP radiosondes. The RS41-SG type of radiosonde measures height using GPS,
and the pressure is calculated with the very same equation.
As discussed in the last subsection, during the descent the radiosonde overestimates air temperature, mostly in the stratosphere,
where the descent speeds are high enough to have an impact. This overestimation of temperature leads to underestimation of
air density. For the RS41-SGP it means that (negative) height increments are smaller than they should be and thus for the
certain pressure level, higher altitudes are reported than they should be. As the height errors accumulate during the descent,
the shift of height still remains in the troposphere levels, where direct heating impact is negligible.
For the RS41-SG radiosondes, the effect is very similar, resulting in an underestimation of pressure increments, causing a
lower pressure for a given height. And vice versa, lower height for the given pressure. Illustration of this effect can be seen in
Figure 23.
The shift of the profile is visible only if we use as a vertical coordinate the variable which is calculated, not directly measured.
As most applications (including many NWP systems) use pressure as vertical coordinate, the effect can be seen for RS41-SG
radiosondes. It should lead to an increase in SD when comparing variables to the NWP model, but also to increase of
tropospheric temperature bias due to the temperature gradient in the troposphere (as can be seen in Finnish data compared to
ECMWF in Figure 14).
The effect is clearly visible in Figure 24. For the Praha-Libuš data sample ascent and descent levels were matched both using
height (blue for bias, red for SD) and using pressure (grey for bias, yellow for SD). In the stratosphere the direct heating of the
sensor has a major effect on T differences and the lines are almost the same. In the troposphere, the friction is much lower due
to the slower DR and for pressure-matched levels, the shift of the profile caused by accumulated pressure errors is responsible
for the majority of bias (difference between gray and blue line). Up to 11 km there is also visible worsening of SD for pressure-
matched profiles due to this effect.





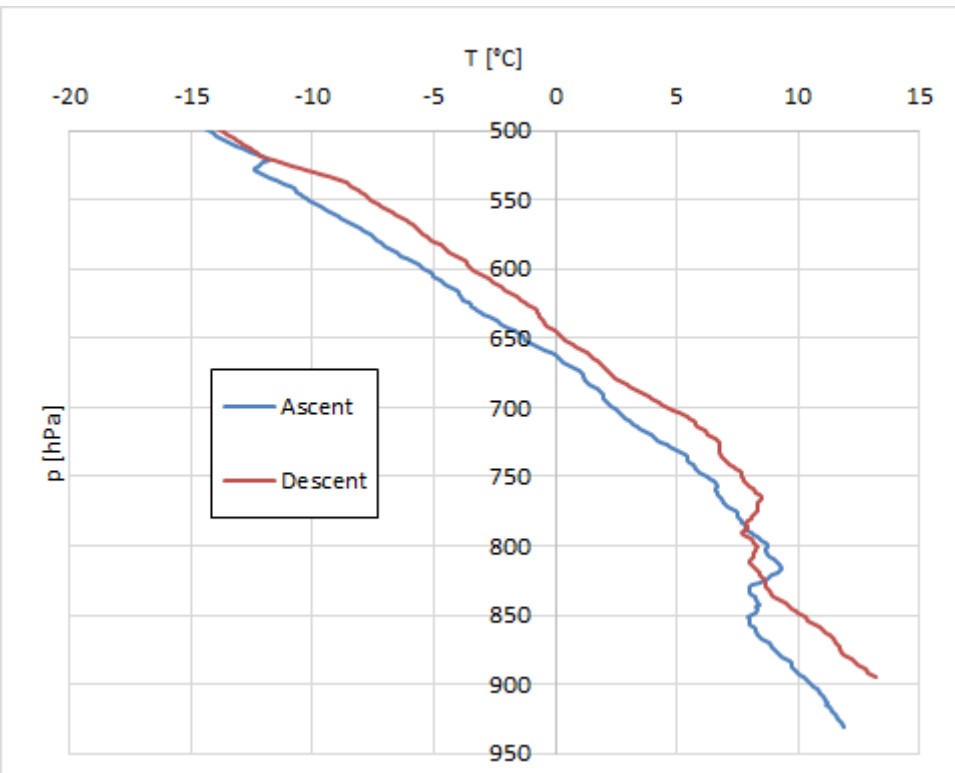


Figure 23: Shift of the tropospheric profile as a function of pressure.  Profile from 23-09-2019, 12  UTC, Praha-Libus.







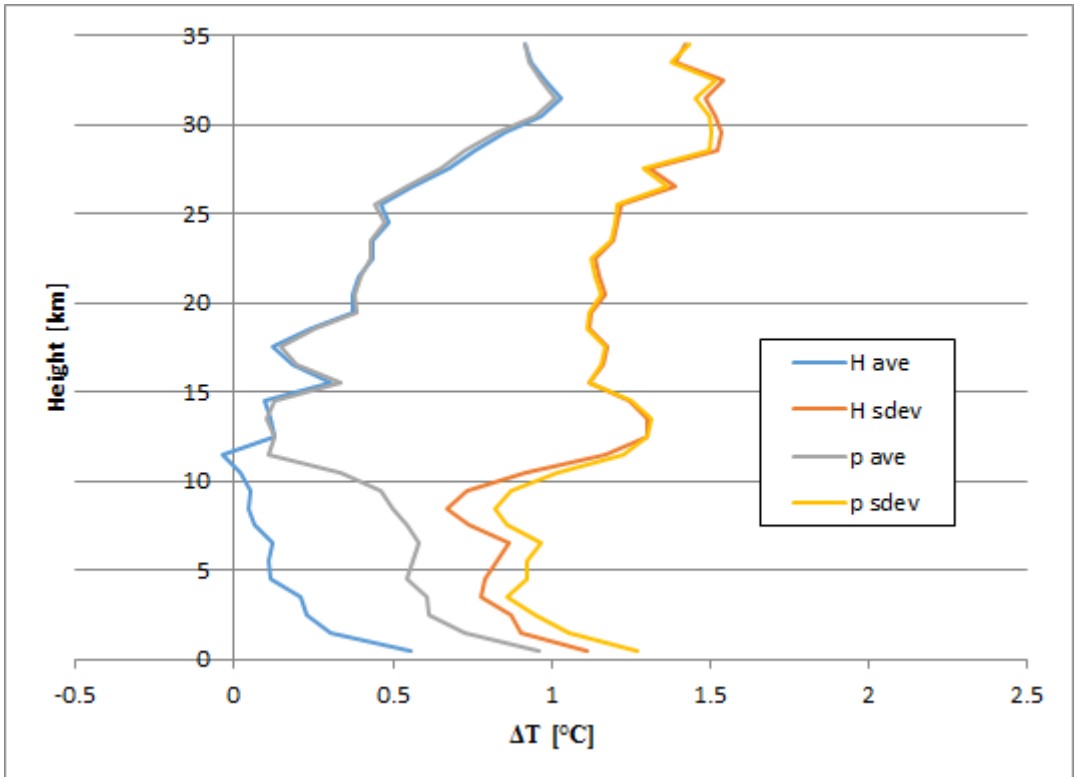


Figure 24: T bias and SD when ascent/descent matched used height (blue, red) or matched using pressure (grey, yellow)


The improvement of pressure differences after application of the temperature correction, Eq (2), and recalculating pressure
using corrected temperature is clearly visible in Figure 25. The recalculation was made on a data sample from Praha-Libus,
and the pressure bias near the surface was decreased by approximately 95 %.




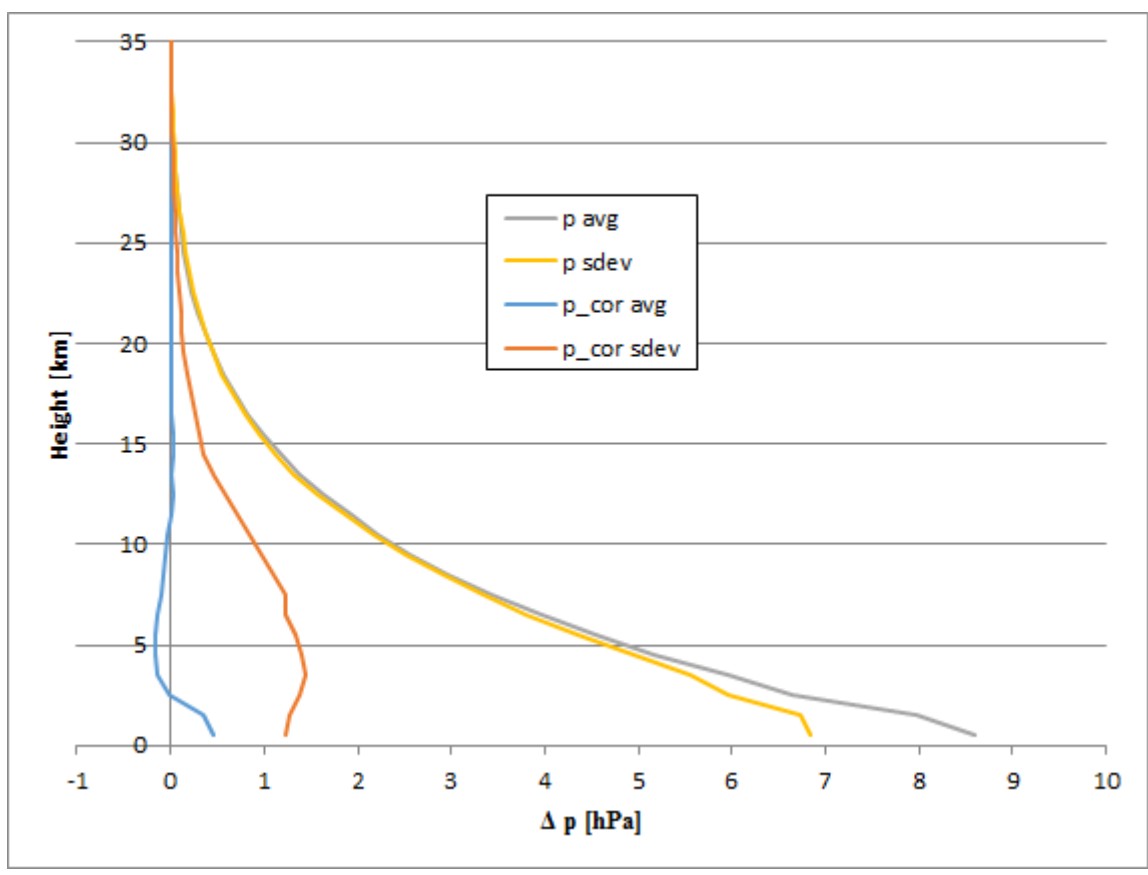

Figure 25: Average Δp and its standard deviation before and after applying temperature correction for friction and pressure recalculation. For representation purposes is $\Delta p = p_A - p_D$ on this plot.

The positive effect of the pressure correction was checked using Praha-Libus temperature data. Figure 26 shows the bias (solid) and SD (dashed) relative to the ascent for the two versions of temperature descent data (all the data used $T_{cor}$ according to Eq. (2)). Green lines are for the data matched by reported pressure and red for the data matched by pressure recalculated using corrected temperature. The negative effect of accumulated pressure error due to friction was removed by the pressure correction. The red lines on figure 26 are almost identical with blue and red lines on figure 22 using data matched by height. Overall, it appears that the pressure errors arising from stratospheric heating of the temperature sensor can largely be removed by using corrected temperatures in the hydrostatic calculations.

481



Figure 26: Temperature bias and SD when as vertical coordinate was used pressure (green) and corrected pressure (red). On all temperatures friction correction was applied.

## 5 Assimilation of descent data

Partly prompted by the drop in numbers of aircraft data due to the Covid-19 pandemic (Ingleby et al, 2021) a trial was run assimilating European RS41 descent data for 20 January to 28 April 2020. The large-scale impact was very small as expected, but over Europe there were modest improvements in the fit of the 12 h forecast to radiosonde ascent data (Figure 27). There were improvements over Germany (not shown) and the impact was mixed over Scandinavia. The decision was taken to assimilate only the German descent data for the time being - this is the best subset, because they have parachutes and pressure sensors, as discussed in section 3 - implemented operationally on 17 June 2020. In the final configuration data from 150 hPa down to the surface were used. Upper-level temperatures were excluded because of the biases. Upper-level winds were used in the trial, but there are some concerns about the pressures being less accurate when the radiosonde is falling fast. The use of



upper-level descent winds may be reviewed in the future. Note that at the upper-levels the ascent and descent are close together
in space and time and so one may not want to assimilate both ascent and descent profiles. As discussed in section 4, some of
the bias problems would be reduced if height was used as the vertical coordinate rather than pressure - however this would
involve significant work and testing, so there are no plans to do so in the near future.

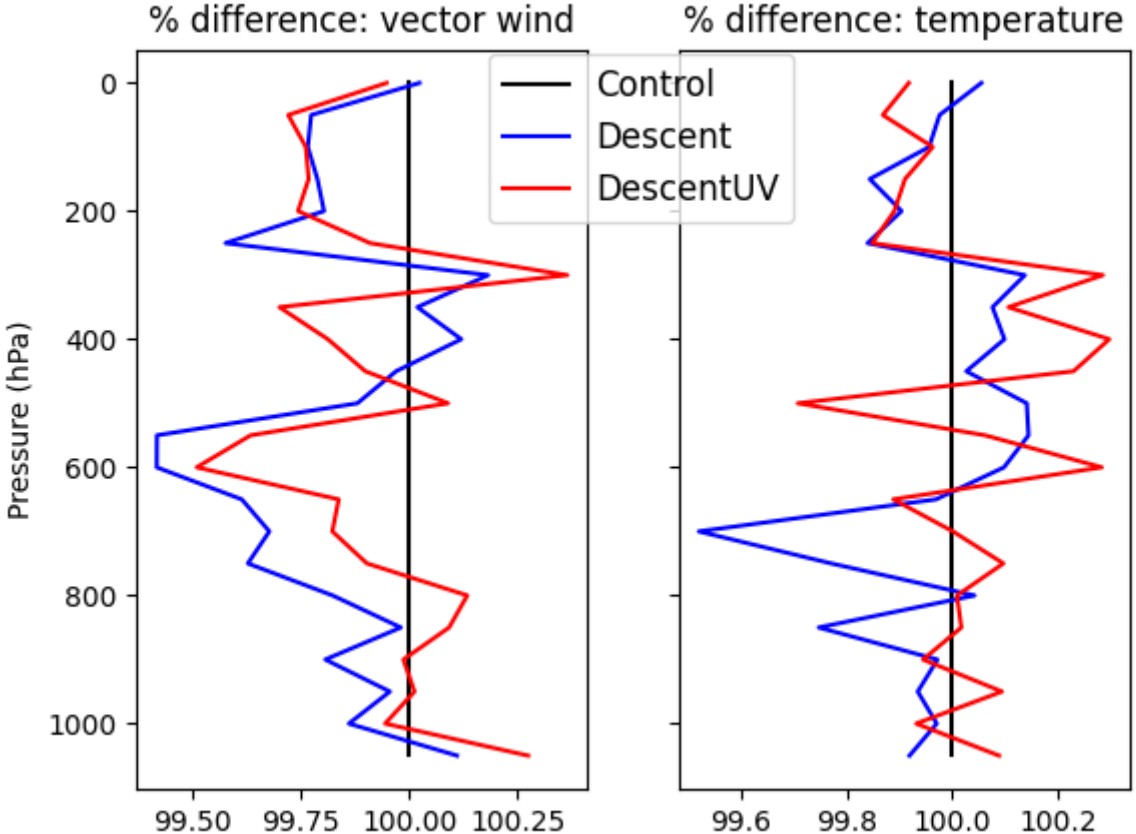


Figure 27. Effect of assimilation of all descent data (wind, temperature, humidity; blue line) or just descent winds (red line)
2020-01-20 to 2020-04-20. Results are shown for temperature and vector wind fit of 12 hour forecasts to European radiosonde
ascents, normalised by the fit of the control forecasts (so values less than 100% indicate improved forecasts).
**6 Discussion and conclusions**
The most obvious difference between ascent and descent data is that the descent temperatures are higher at upper levels. This
had been noted before for different radiosonde types, by Tiefenau and Gebbeken (1989) and Vencat Ratnam et al. (2014). The
latter did not discuss the cause but Tiefenau and Gebbeken (1989) took the descent temperatures as accurate and suggested





that the ascent temperatures were too low due to sampling the balloon wake and adiabatic cooling of the gas within the balloon. Whilst wake effects cannot be completely discounted, our results suggest that the descent temperatures are too high and that this is closely linked to the descent rate (Figure 16). In Figure 21 the dependence on descent rate appears quadratic. Vaisala are working on updated processing to address the temperature bias and other issues. There has been considerable discussion on the source of the ascent/descent temperature differences. Whilst we cannot definitively explain the heating mechanism a plausible hypothesis is a conversion of kinetic energy via frictional heating. Clearly the falling radiosonde (plus balloon remains and parachute if fitted) are slowed by friction otherwise it would accelerate to much higher speeds during the descent. Future work might include testing radiosonde sensors in a wind tunnel with a flow of 20 m/s or more to see if the heating is replicated (care would be needed with the reference temperature). We have not been able to find such tests in the literature. There is a paper by de Podesta et al. (2018) about the effect of sensor diameter on temperature errors but they were looking at lower flow rates.

Another difference, that doesn't seem to have been reported before, is that on average the descent winds are smoother than the ascent winds. In part this seems to reflect the fact that ascents are generally more affected by pendulum motion, however inertial effects and the filtering applied to 'remove' pendulum motion also play a role. The smoother descent winds have a closer RMS fit to the NWP winds, but we cannot currently say whether the ascent or descent winds are more accurate. Most studies of radiosondes concentrate on the temperatures and humidities and the winds are somewhat neglected; the use of long strings improves stratospheric temperatures at the expense of increasing the pendulum motions. For aircraft the winds have more than twice the impact of the temperatures on the quality of short-range forecasts (Ingleby et al, 2021) and forecast sensitivity diagnostics suggest that the same is true of radiosondes (Pauley and Ingleby, 2021), partly because satellite instruments primarily provide temperature and humidity data. Experience shows that GPS winds are generally good quality and biases do not seem to be a problem. GPS can provide high vertical resolution winds - but this makes pendulum motion more obvious and avoidance or removal of pendulum motion deserves more attention. Sako and Walterscheid (2016) discuss empirical filtering of wind profiles from radiosondes and Jimsphere balloons used specifically for wind measurement. It seems likely that dropsonde wind profiles are closer to the true winds (Wang et al, 2008) than radiosonde ascent or descent winds. The two balloon ascents of Kräuchi et al. (2016) largely eliminate pendulum motion but need more evaluation.

Some aspects of the descent data can be improved by estimating and removing heating effects due to high fall rates (on the temperature, and also on the pressure for radiosondes without a pressure sensor). The descent characteristics are more variable than ascent rates in that for balloons with parachutes, the manner in which the parachute deploys can affect the amplitude of the pendulum motion and the descent speed. It is also likely that there can be improvements in the filtering of pendulum motion. Vaisala are working on these aspects but are not yet ready to give a timescale for changes. In principle users could apply bias corrections, but improving the winds is difficult if they have already been filtered. On the whole, it is simplest to stick to the current practice of manufacturers providing best estimate profiles, but more details of the processing would be





welcome and this area should be kept under review. On a similar note there is a question of whether there should be a GRUAN
descent product for the RS41 - more work on the uncertainties would be needed for this. There is the wider question of how
much the lessons learnt from the RS41 descent are applicable to other radiosondes such as the Meteomodem M10. There is
some evidence that pressure sensor accuracy is worse whilst falling fast, but more work on this is needed. However the fall
speed should have very little effect on the accuracy of GPS derived positions, because the GPS satellites are moving much
faster anyway.

There is evidence that use of parachutes and/or pressure sensors gives some improvement to the descent data (this will reduce
with improved processing/bias correction). There is also the possibility of installing extra receivers so as to obtain more descent
data from the lower troposphere (this has been demonstrated in Corsica, Peyrat, pers. comm. 2020). Whether the extra costs
are worthwhile would need to be assessed. We note that the impact of extra radiosonde profiles over well-observed Europe
will be less than the impact of extra profiles near remote islands or ships. In May 2021 descent data was received from several
European ships in the North Atlantic and also a station in Antarctica. ECMWF and DWD have started operational assimilation
of a subset of descent profiles - excluding the stratospheric segment with higher average fall rates (arguably it would be better
to exclude data based on the actual descent rate, but this is not reported, it is also desirable to exclude values where there are
particularly large accelerations, e.g. Fig. 5). The US Navy global model is assimilating all available descent profiles (Pauley,
pers. comm. 2021), we are not aware of other NWP systems using them yet. NWP systems generally use pressure as the vertical
coordinate for radiosonde data, arguably there would be advantages in using height instead. There has been much more use of
NWP model fields in this investigation than is traditional for development/validation of in situ observations (but routine now
for new satellites - Newman et al. 2020). This means that a much larger sample can be examined. Note that traditional
radiosonde intercomparisons (e.g., Nash et al, 2011) can't be used to assess descent data because the multi-radiosonde rig used
has various implications for the descent including possible entanglement.
**7 Appendix: comparison with aircraft temperatures**
For cruise level aircraft (typical speed is about 250 m/s) the measured temperature, known as total air temperature (TAT), can
be more than 20 K higher than the static air temperature (SAT). The link between TAT and SAT (Wendisch and Brenguier,
2013) can be expressed with equation:
$$SAT = \frac{TAT}{1+\frac{r(\gamma-1)}{2}\cdot M^2} \quad [A1],$$
where $r$ is the recovery factor of the sensor, $\gamma$ adiabatic index and $M$ is Mach number.
From equation [A1] we can get the difference $TAT - SAT = SAT \cdot \frac{r(\gamma-1)}{2} \cdot M^2$
If we use the Mach number: $M = \frac{v}{a}$,



where $v$ is the airspeed of the object (aircraft or radiosonde) and $a$ the speed of sound, given by: $a = \sqrt{\gamma \cdot (C_p - C_v) \cdot SAT}$,
where $C_p$ and $C_v$ are heat capacity constants for constant pressure and volume respectively,
we can get: $TAT - SAT = \dfrac{r(\gamma-1)}{2} \cdot \dfrac{v^2}{\gamma \cdot (C_p - C_v)}$ [A2]
Applying $\gamma = \dfrac{C_p}{C_v}$ to equation [A2], the difference between measured and real temperature is:
$$TAT - SAT = \frac{v^2}{2C_p} r \qquad\qquad \text{[A3]},$$
According to the Wikipedia entry on Total air temperature, the typical recovery factor for platinum wire (which is used for
radiosondes) is 0.75 – 0.9. Even if we are not sure about the exact physical process of kinetic energy transfer to internal heat
in case of radiosondes, we might expect similar behavior (quadratic dependency on descent rate, but independent of height,
temperature and air density). When we apply the equation A3 to descending radiosondes, where $v$ is the descent rate ($DR$) of
the radiosonde, with the range 0.75 - 0.9 for $r$ we get:
$TAT - SAT = A \cdot 10^{-4} \cdot DR^2$   with A in the range 3.70 to 4.47.     [A4]
However, the similar coefficients for aircraft and descent radiosondes may come from different conversion mechanisms..

**Data availability statement**
The radiosonde descent data for September-November 2019 are available in BUFR format from
https://www.gruan.org/data/data-packages/dpkg-2021-2. Radiosonde ascent data in BUFR format are available from
https://www.ncei.noaa.gov/pub/data/igra/v1/related/BUFR/ecmwf/data/ (see Geller et al, 2021). BUFR decoding tools are
available from ecCodes (with examples of radiosonde decoders at https://confluence.ecmwf.int/display/ECC/bufr_read_tempf
). Other decoders are also available, e.g. https://github.com/NOAA-EMC/NCEPLIBS-bufr/tree/master .
**Author contributions**
BI with help from DE worked on paper conceptualization and on the methodology. BI, MM and GM performed the formal
analysis and visualisation. All the co-authors contributed to the writing of the paper, review and editing. Data curation was by
MS, BI and DE.
**Competing interests**
The authors declare that they have no conflict of interest.
**Acknowledgements**
The authors would like to thank the meteorological services that provided, and continue to provide, descent data. Michael de
Podesta (NPL) gave useful advice on the measurement of temperature by a moving sensor. Some early work on descent data



was performed by Christopher Wyburn-Powell as a vacation student at the Met Office in summer 2018. Lars Isaksen and Sean
Healy of ECMWF made suggestions that improved the manuscript.

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
