# Peer review of "On the quality of RS41 radiosonde descent data"

_Atmospheric Measurement Techniques, 2021_

## Referee Comment (RC1)

**Review on 'On the quality of RS41 radiosonde descent data' by Bruce Ingleby et al., (ATM-2021-183).**

This paper presents detailed analysis carried out to test the quality of the descent data obtained from RS41 radiosonde over several stations from Europe. Initially they compared between ascent and descent profiles of all the meteorological variables (T, RH, U and V) and later with independent ECMWF short-range forecasts and also radio occultation profiles. Difference between ascent and descent profiles has been attributed majorly due to pendulum motion besides large terminal velocities. Finally, it was concluded that descent data is much smoother and can be used for data assimilation in NWP model, that can be obtained (descent) with NO additional cost. Overall, the manuscript is well written and sound enough both technically and scientifically. The main topic of research is worth investigating and fits well within the scope of Atmospheric Measurement Techniques (AMT) Journal. However, there are several aspects that remain unclear at this stage and I recommend for its acceptance only after taking care of the following major and minor comments/suggestions.

**Major comments/suggestions:**

1.  Authors have missed one important aspect of estimating the descent rates using balance between gravity and drag forces by considering actual dimensions and weight of RS41. This has important consequences on the measurements of radiosonde during the descent. It does not matter whether a parachute is attached or not. It purely depends on the amount of weight left with the busted balloon. If more weight is left than the radiosonde weight, smooth profile can be expected.

2.  Similar study was made long back by Venkat Ratnam et al., (2014) where they assed radiosonde descent data from a tropical region. Not much focus is given to this original research and mentions many places in the current manuscript in such a way that this analysis is made for the first time. Proper credit should be given to the original research at many places throughout the manuscript.

3.  Initially as soon as the balloon is busted, the terminal (fall) velocity increases drastically and it takes some time to stabilize. Since density at those altitudes are very low, a busted balloon will have much higher velocities and as density increases at lower altitudes it slows down. Thus, previous works have recommended to use data from 5-10 km below from the busted height. I strongly feel the same thing will be valid in the current works also. Authors should explicitly discuss this aspect.

4.  When the busted balloon weight is high, it will not allow the radiosonde to drift freely with the background wind. Instead it drags the radiosonde. In this case, whether wind velocities measured by the radiosonde during its descent will be realisable? In this case note that wind velocities are not from a freely floating body. Same problem may arise even for temperature and humidity when the response time is not good enough. When a busted balloon is at high velocities (close to 80-100 m/s, in Figure 5,6 and 7), whether any sensor (Particularly T and RH) in the radiosonde is capable of sensing the background atmosphere? I suggest providing details on these aspects in the manuscript.

5. Unable to see big difference in T (Figure 14 and 15) and RH (Figures 18 and 19) between ascent and descent profiles over any country. Then the question arises what will be its impact in NWP if assimilated? Again later, I am surprised to see the effect of assimilation of all the descent data you have seen here (Figure 27). I am unable to see a big difference In T and RH between ascent and decent profiles. Then from where this effect has come?

6. I suggest authors show the diurnal variation in all the meteorological variables first using ECMWF data before making any conclusive statement (Lines 381-385 also lines 389-391.). If the variation is much higher than the expected diurnal variation, then only one should talk about bias. Diurnal variation is too small over higher latitudes hence big difference is not expected in the ascent and descent profiles. However, the situation will be completely different over tropical or equatorial latitudes. In fact, similar exercise needs to be done over low latitudes before making any conclusive statements.

**Specific comments/suggestions:**

1. Lines 28-29. This was first reported by Venkat Ratnam et al., (2014).

2. Lines 40-41. As soon as the balloon bursts, it will come much faster, similar to the release of a satellite from a rocket. It cannot be much faster whether one uses a parachute or not. It is to be noted that the role of the parachute comes only after stabilising that may take about 5-10 km below from the busted altitude.

3. Lines 50-51. It is rightly mentioned that a balloon will be advected horizontally by the background wind and typically travels 50-300 km…In fact drift depends on the background winds and will be different in regions of the world. Since most of the stations used in this manuscript fall under mid-latitudes, balloon drift mostly depends on the wind speed of sub-tropical jets.  I suggest plotting the balloon drift something similar to Figure 3 and 4 of Venkat Ratnam et al., (2014) season wise and discuss the same at relevant places.

4. From Figure 2 it is not clear out of how many profiles, those descent profiles were obtained.

5. Lines 83-84. Upper part of the descent is close to the upper part of the ascent in both time and space. Then what about the role of large fall velocities?

6. Lines 108-111. It is mentioned that 'On ascent the sensor boom projects above the radiosonde, so that it samples air that has not flowed over the body of the radiosonde. On descent, with a working parachute, it should be in a similar position - so it may sample air that has flowed over the radiosonde body, which has the potential to introduce biases or contamination. It is not known how a radiosonde descending without a parachute is orientated, or if it may be tumbling.' I do not fully agree with these statements. The orientation of the radiosonde fully depends on the weight of the busted balloon. If the weight is much higher than the weight of the radiosonde, it is quite common to think that the radiosonde will be dragged with the busted balloon coming down first.

7. Lines 123-125. Again, it depends on the weight of the busted balloon?

8.  Lines 151-152. Again, it depends on the weight of the busted balloon? This reason is already suggested by Venkat Ratnam et al., (2014).

9.  Figure 7. It looks like there is a latitudinal effect on the descent rates that is shown in figure. Can you check the effect of background temperatures/densities at those latitudes?

10. Lines 196-197. It also has strong seasonal variation. I suggest plotting the maximum drifts at all the stations to make a conclusive statement.

11. Lines 229-230. It depends on the weight of the left-out balloon after the burst.

12. Figures 9b and 10b. It is not the vertical velocity. Vertical velocity has a different meaning in meteorology. Suggest replacing it with 'descent speed' or something similar.

13. 259-260. I do not fully agree with this statement. It depends on the left-out balloon weight after it busted and does not depend on whether the parachute is attached or not.

14. Line 269. Your ECMWF forecast is at 9 km horizontal grid spacing. How is the comparison done when the balloon drift is too high (>300 km) which changes with the altitude?

15. Lines 283-285. It is mentioned that 'One surprise was that the descent profiles (in red) fit the background more closely than the ascent profiles (in black), particularly at upper levels. Comparing individual ascent/descent profiles the descent winds generally appear smoother and this appears to be the cause of the better fit to background.' First of all, I am unable to see smoother profiles as mentioned. It can be due to descent rate being not affected while using GPS satellites which move much faster. Also, both time (within a few minutes) and space (sensing the same background) is very low, so no variability can be noticed. Descent profiles look much smoother because it is not freely floating but dragged instead? I suggest providing details on each of these aspects in the revised version.

16. Line 310. Cool by 0.5°C. I suppose this is within the uncertainty of measurement?

17. Line 314-315. (also Table 2). I do not understand what do mean in these lines. Note that RO is limb technique and there will be a maximum difference of 300 km from the top of the profile (~40 km) to the surface. The statement 'the RO data is much closer to the ascent temperatures then the descent temperature' is not correct. It depends on whether you are considering rising or setting occultation. You should consider the mean latitude and longitude of the total profile in RO data and then check whether it is within the limits of selection criteria (within $1^0$ or 100 km).

18. Lines 331-332. In the previous section, you have mentioned that the time and space difference at upper levels will be minimum so that a good comparison is seen. But in these lines, you are mentioning that the large top-level ascent-descent difference has disappeared by 300 hPa which is in contradiction. Be specific here. Further, offset of 0.2 or 0.3°, is it not this is within the uncertainty of the measurement? If not, what about temporal variation? Have you checked whether this offset is positive or negative at other

timings? I suppose it becomes negative if we consider evening profiles rather than morning profiles.

19. In Figures 15 and 16. I suppose the large difference in T at higher descent rates may be due to sensor problems (Unable to sense that fast?).

20. Lines 362-364. These statements may not be true if you make a comparison during day and night times. Once should consider expected diurnal variation before making such statements.

21. Line 365-366. I think the bias can also be due to sensor response. In one second, radiosonde travelled almost 100 m in which background temperature may not be the same. We need a high response sensor to check this.

22. Lines 493-494. Even if someone wants to assimilate, you have already reported higher differences in all the parameters due to large descent speeds?

23. Line 502-503. Very late these references (check the spelling of Venkat) have appeared in this paper. I suggest giving proper credit to the original works being carried on this topic. These references should be very well discussed in the introduction section itself. Further the statement ' ..latter did not discuss the cause……' is not correct. It was mentioned very clearly that due to large descent rates, sensors do not respond that quickly to make meaningful measurements. Further, one cannot combine these two references. Note that the latter reference already attributed differences due to the diurnal variation. Several reasons are mentioned in the Venkat Ratnam et al., (2014) and I suggest authors to go through the paper very carefully before making such statements.

24. Lines 517-518. I do not agree with these statements. Note that descent winds are not from free floating of the balloon but dragged due to acceleration due to gravity in addition to the background wind.

25. Lines 519-520. One cannot think of having accurate measurement of wind during descent as it is dragged by acceleration due to gravity that depends on left out busted balloon and weight of radiosonde in addition to the background wind.

26. Line 521. Here references are needed for those studies. Also, I guess, those studies have been neglected due to known reasons (dragged winds rather than free floating along with background).

27. Lines 525-527. Can one think of having a good quality balloon (almost sphere rather than oval after filling the gas) to make the balloon stable?

28. Line 537. I doubt the descent winds for the reasons mentioned above.

29. Lines 539-540. I strongly suggest having a GRUAN descent product for the RS41 (for that matter any radiosonde type) as one more profile is being obtained at no cost. This is very

useful particularly launched in the tropical altitudes as the expected diurnal variation is very high.

30. Line 542. Same holds good even for T. Check the response time of the temperature sensor at higher descent speeds.

31. Line 546. I suggest providing a reference for this statement. Also, I do not agree that using parachutes will give some improvement.

32. I suggest to smooth the data with 100 m to take out the random motion of the balloon unless there is a need for very high vertical resolution. Further, 1s sampling particularly at higher altitudes (lower densities) may not be sufficient to sample the background atmosphere. I do not think that any NWP model requires parameters with very high vertical resolution for assimilation.

**References:**

Venkat Ratnam, M., N. Pravallika, S. Ravindra Babu, G. Basha, M. Pramitha, and B. V. Krishna Murthy (2014), Assessment of GPS radiosonde descent data, Atmos. Meas. Tech., 7, 1011–1025.

---x----

---

## Author Comment (AC2)

Authors' response

RC1. Review on 'On the quality of RS41 radiosonde descent data' by Bruce Ingleby et al., (AMT-2021-183).

This paper presents detailed analysis carried out to test the quality of the descent data obtained from RS41 radiosonde over several stations from Europe. Initially they compared between ascent and descent profiles of all the meteorological variables (T, RH, U and V) and later with independent ECMWF short-range forecasts and also radio occultation profiles.

Difference between ascent and descent profiles has been attributed majorly due to pendulum motion besides large terminal velocities. Finally, it was concluded that descent data is much smoother and can be used for data assimilation in NWP model, that can be obtained (descent) with NO additional cost. Overall, the manuscript is well written and sound enough both technically and scientifically. The main topic of research is worth investigating and fits well within the scope of Atmospheric Measurement Techniques (AMT) Journal. However, there are several aspects that remain unclear at this stage and I recommend for its acceptance only after taking care of the following major and minor comments/suggestions.

**Thank you. On the issue of cost, there is a minor cost from keeping the ground station operating during the descent. If there are changes specifically for the descent data (adding a parachute, a pressure sensor or an additional receiving station) the cost is increased. Hence, we prefer to say that the additional cost is small (rather than zero).**

Major comments/suggestions:

1. Authors have missed one important aspect of estimating the descent rates using balance between gravity and drag forces by considering actual dimensions and weight of RS41. This has important consequences on the measurements of radiosonde during the descent. It does not matter whether a parachute is attached or not. It purely depends on the amount of weight left with the busted balloon. If more weight is left than the radiosonde weight, smooth profile can be expected.

**The descent rate is important, we make that clear. We don't understand "It does not matter whether a parachute is attached or not." because a parachute does slow the descent. We now mention the descent rate estimates of Venkat Ratnam et al., (2014), but because of the unknowns (balloon mass present, orientation) such estimates seem to give an upper bound for the descent rate (figure 6 of VR14). Because we have the measured descent rates having such an estimate does not add much.**

2. Similar study was made long back by Venkat Ratnam et al., (2014) where they assed radiosonde descent data from a tropical region. Not much focus is given to this original research and mentions many places in the current manuscript in such a way that this analysis is made for the first time. Proper credit should be given to the original research at many places throughout the manuscript.

**We now discuss the two previous studies in the introduction and provide more details of Venkat Ratnam et al., (2014) [VR14] in particular, we also added a mention in section 2.4.**

3. Initially as soon as the balloon is busted, the terminal (fall) velocity increases drastically and it takes some time to stabilize. Since density at those altitudes are very low, a busted balloon will have much higher velocities and as density increases at lower altitudes it slows down. Thus, previous works have recommended to use data from 5-10 km below from the busted height. I strongly feel the same thing will be valid in the current works also. Authors should explicitly discuss this aspect.

**At the end of VR14 it says "Thus, the data within a few kilometers from the balloon burst may be biased because of improper sampling." (which we now quote). Rather than restrictions based on height relative to balloon burst it seems better to use the fall rate (but this is not currently available within the ECMWF NWP system, so a simple pressure limit is used). At the end of the paper we have added "Descent data should be used with caution and sections with high descent rates particularly so, however different users have different tolerances and we expect that improved processing will increase the proportion of usable data."**

4. When the busted balloon weight is high, it will not allow the radiosonde to drift freely with the background wind. Instead it drags the radiosonde. In this case, whether wind velocities measured by the radiosonde during its descent will be realisable? In this case note that wind velocities are not from a freely floating body. Same problem may arise even for temperature and humidity when the response time is not good enough. When a busted balloon is at high velocities (close to 80-100 m/s, in Figure 5,6 and 7), whether any sensor (Particularly T and RH) in the radiosonde is capable of sensing the background atmosphere? I suggest providing details on these aspects in the manuscript.

**It is unclear if the reviewer is thinking of the balloon remains as a compact ball or as causing drag like a parachute (the latter seems more likely). Regarding the winds we mention "an 'inertial' correction for the delayed response to horizontal wind shear (Appendix of Hock and Franklin, 1999)" – not currently applied in MW41 processing. Stratospheric radiosonde humidities are not assimilated anyway. We have added notes on temperature sensor response times to section 3.3.**

5. Unable to see big difference in T (Figure 14 and 15) and RH (Figures 18 and 19) between ascent and descent profiles over any country. Then the question arises what will be its impact in NWP if assimilated? Again later, I am surprised to see the effect of assimilation of all the descent data you have seen here (Figure 27). I am unable to see a big difference In T and RH between ascent and decent profiles. Then from where this effect has come?

**One of the main points of Figure 14 and 18 is that (apart from temperature biases) the descent and ascent quality are similar – so we have more usable profiles. Because the ascent and descent positions/times differ they have different background (B) profiles.**

6. I suggest authors show the diurnal variation in all the meteorological variables first using ECMWF data before making any conclusive statement (Lines 381-385 also lines 389-391.). If the variation is much higher than the expected diurnal variation, then only one should talk about bias. Diurnal variation is too small over higher latitudes hence big difference is not expected in the ascent and descent profiles. However, the situation will be completely different over tropical or equatorial latitudes. In fact, similar exercise needs to be done over low latitudes before making any conclusive statements.

**In the ECMWF O-B statistics the B values take the (background) diurnal variation into account whereas in section 4 the Praha ascent-descent statistics (those discussed in lines 381-391) will sample the diurnal cycle. Figures 16 and 17 show some ECMWF O-B statistics by time of day (colour coding). In practice ECMWF O-B statistics show no clear diurnal variation (example below). We agree that it would be good to see similar results for low latitudes, but do not expect tropical results to be "completely different".**

**I have looked at figures like the one below - showing 00 and 12 UTC statistics separately for Germany and Norway, note that some points have small samples - and concluded that there is no clear diurnal cycle in the O-B statistics. It is normal to briefly report a result like this rather than present lots of figures.**

[Figure]

**Specific comments/suggestions:**

1. Lines 28-29. This was first reported by Venkat Ratnam et al., (2014).

**Venkat Ratnam et al., (2014) is now mentioned in the introduction.**

2. Lines 40-41. As soon as the balloon bursts, it will come much faster, similar to the release of a satellite from a rocket. It cannot be much faster whether one uses a parachute or not. It is to be noted that the role of the parachute comes only after stabilising that may take about 5-10 km below from the busted altitude.

**Figures 9 and 10 plus S4 and S5 suggest that sometimes the parachute opens suddenly after falling fast and sometimes it seems moderately effective immediately after balloon burst.**

3. Lines 50-51. It is rightly mentioned that a balloon will be advected horizontally by the background wind and typically travels 50-300 km...In fact drift depends on the background winds and will be different in regions of the world. Since most of the stations used in this manuscript fall under mid-latitudes, balloon drift mostly depends on the wind speed of sub-tropical jets. I suggest plotting the balloon drift something similar to Figure 3 and 4 of Venkat Ratnam et al., (2014) season wise and discuss the same at relevant places.

**Given that the paper is already long (noted by reviewer 3) and that the manuscript is not about balloon drift per se we do not propose to add any more figures on drift – an indication is given in Figure 1.**

4. From Figure 2 it is not clear out of how many profiles, those descent profiles were obtained.

**I think the question is about the ratio of the number of ascent profiles to descent profiles. The last sentence of the paragraph has been modified to read 'Some ascents, usually less than 5%, do not have a corresponding descent report, …'.**

5. Lines 83-84. Upper part of the descent is close to the upper part of the ascent in both time and space. Then what about the role of large fall velocities?

**Large fall velocities are dealt with later in sections 3 and 4.**

6. Lines 108-111. It is mentioned that 'On ascent the sensor boom projects above the radiosonde, so that it samples air that has not flowed over the body of the radiosonde. On descent, with a working parachute, it should be in a similar position - so it may sample air that has flowed over the radiosonde body, which has the potential to introduce biases or contamination. It is not known how a radiosonde descending without a parachute is orientated, or if it may be tumbling.' I do not fully agree with these statements. The orientation of the radiosonde fully depends on the weight of the busted balloon. If the weight is much higher than the weight of the radiosonde, it is quite common to think that the radiosonde will be dragged with the busted balloon coming down first.

**This seem to assume that the balloon remains form a compact (heavy) ball, but it is also plausible that they form strips – waving above and slowing the descent (see 8 below). Arguably figure 6 of VR14 suggests that remains trailing above are more common than a ball below. It seems reasonable to say simply that we do not know the orientation.**

7. Lines 123-125. Again, it depends on the weight of the busted balloon?

**These lines about pendulum motion are primarily about the ascent.**

8. Lines 151-152. Again, it depends on the weight of the busted balloon? This reason is already suggested by Venkat Ratnam et al., (2014).

**Added 'Venkat Ratnam et al. (2014) suggested that the balloon remains sometimes act as a parachute.'**

9. Figure 7. It looks like there is a latitudinal effect on the descent rates that is shown in figure. Can you check the effect of background temperatures/densities at those latitudes?

**As we say 'The Norwegian radiosondes fall faster than those from the other countries studied - it is unclear why they fall faster than the Finnish radiosondes.' We tried asking radiosonde experts**

from these countries but no clear explanation was available (it could relate to the type of balloon used, or whether the radiosondes were 'hardshell' or the newer, lighter 'softshell' type). We think this is a question for a later study – ideally with some tropical data.

10. Lines 196-197. It also has strong seasonal variation. I suggest plotting the maximum drifts at all the stations to make a conclusive statement.

**Not in the scope of this study.**

11. Lines 229-230. It depends on the weight of the left-out balloon after the burst.

**The main difference between the fall rates in figures 9 and 10 appears to be due to the parachute, which we mention.**

12. Figures 9b and 10b. It is not the vertical velocity. Vertical velocity has a different meaning in meteorology. Suggest replacing it with 'descent speed' or something similar.

**Changed to 'descent rate' for consistency.**

13. 259-260. I do not fully agree with this statement. It depends on the left-out balloon weight after it busted and does not depend on whether the parachute is attached or not.

**The statement is: "The motion on descent may be more consistent if the radiosonde could be 'cut free' of the balloon remains and fall on its own without a parachute." We think this is very likely (if the balloon remains are small and there is no parachute then cutting the string would make little difference). The balloon remains are mentioned at various points in the text.**

14. Line 269. Your ECMWF forecast is at 9 km horizontal grid spacing. How is the comparison done when the balloon drift is too high (>300 km) which changes with the altitude?

**This is described a few lines further down ('and in this study' added as clarification): "Afterwards, and in this study, ascent profiles were split into sub-profiles of 15-minutes each and treated as valid at the time and latitude/longitude of the first point in the sub-profile. Descent profiles are split into 5-minute sub-profiles for comparison with the model."**

15. Lines 283-285. It is mentioned that 'One surprise was that the descent profiles (in red) fit the background more closely than the ascent profiles (in black), particularly at upper levels. Comparing individual ascent/descent profiles the descent winds generally appear smoother and this appears to be the cause of the better fit to background.' First of all, I am unable to see smoother profiles as mentioned. It can be due to descent rate being not affected while using GPS satellites which move much faster. Also, both time (within a few minutes) and space (sensing the same background) is very low, so no variability can be noticed. Descent profiles look much smoother because it is not freely floating but dragged instead? I suggest providing details on each of these aspects in the revised version.

**Start of next sentence changed to 'This is illustrated in Figure 12 which …'. I have added plots of four ascent/descent pairs from Lindenberg to the supplement to provide examples.**

16. Line 310. Cool by 0.5°C. I suppose this is within the uncertainty of measurement?

**In the text: "at about 50 hPa, in the extratropics, the ECMWF background is too cool by about 0.5°C (this can be seen against the RS41 ascent data in Figure 14)." There is other evidence to support this - see Shepherd et al (2018) referenced in the next sentence.**

17. Line 314-315. (also Table 2). I do not understand what do mean in these lines. Note that RO is limb technique and there will be a maximum difference of 300 km from the top of the profile (~40 km) to the surface. The statement 'the RO data is much closer to the ascent temperatures then the descent temperature' is not correct. It depends on whether you are considering rising or setting occultation. You should consider the mean latitude and longitude of the total profile in RO data and then check whether it is within the limits of selection criteria (within 10 or 100 km).

**These sentences have been rewritten: "More recent work on the analysis system has approximately halved the short-range forecast bias (Laloyaux et al., 2020); they included comparison against radio occultation (RO) retrievals. We compared radiosonde ascent/descent pairs with RO retrievals that were within 100 km and 2 hours of the burst point. The RO data is much closer to the ascent temperatures than the descent temperatures - note that the sample size is much smaller than for the O-B statistics."**

**The RO data comes with a single (average) location and we are comparing with the burst location (relevant for upper levels; the 300 km in the comment seems to refer to drift from the launch point, it should be clearer now that we mention the burst point).**

18. Lines 331-332. In the previous section, you have mentioned that the time and space difference at upper levels will be minimum so that a good comparison is seen. But in these lines, you are mentioning that the large top-level ascent-descent difference has disappeared by 300 hPa which is in contradiction. Be specific here. Further, offset of 0.2 or 0.3°, is it not this is within the uncertainty of the measurement? If not, what about temporal variation? Have you checked whether this offset is positive or negative at other timings? I suppose it becomes negative if we consider evening profiles rather than morning profiles.

**As stated above, because the ascent and descent positions/times differ they have different background (B) profiles and the B values include (the model version of) the diurnal cycle. The upper level bias is the 'direct effect of heating', radiosondes without pressure sensors (Finland, UK) suffer from the 'indirect effect of heating' in the troposphere. As we say these issues are 'discussed in more detail in the next section'. [Example 00Z and 12Z statistics are shown above.]**

**A difference of 0.2° would be within the uncertainty of an individual measurement. Here we are looking at an average over many profiles and a crucial but difficult issue is whether/how much the errors are correlated between different profiles/days/stations. Assuming that the errors are partly random the uncertainty in a long-term average will be smaller than that in an individual profile.**

19. In Figures 15 and 16. I suppose the large difference in T at higher descent rates may be due to sensor problems (Unable to sense that fast?).

**Notes on the response time of the temperature sensor have been added and a reference to the Vaisala RS41 white paper. A time lag in the sensor can't cause the systematic difference in the stratosphere, as there is almost no vertical gradient there over the three-month average. We also checked and found that implementing a 1-3 s shift in the profile makes the results worse, not better.**

20. Lines 362-364. These statements may not be true if you make a comparison during day and night times. Once should consider expected diurnal variation before making such statements.

**As stated above (response to major comment 6), the diurnal variation was examined.**

21. Line 365-366. I think the bias can also be due to sensor response. In one second, radiosonde travelled almost 100 m in which background temperature may not be the same. We need a high response sensor to check this.

**See response to 19.**

22. Lines 493-494. Even if someone wants to assimilate, you have already reported higher differences in all the parameters due to large descent speeds?

**The text was rewritten and simplified: "Upper-level winds were used in the trial, but because of concerns about accuracy when the radiosonde is falling fast all descent data with pressure less than 150 hPa are excluded in the current operational system."**

23. Line 502-503. Very late these references (check the spelling of Venkat) have appeared in this paper. I suggest giving proper credit to the original works being carried on this topic.

These references should be very well discussed in the introduction section itself. Further the statement ' ..latter did not discuss the cause......' is not correct. It was mentioned very clearly that due to large descent rates, sensors do not respond that quickly to make meaningful measurements. Further, one cannot combine these two references. Note that the latter reference already attributed differences due to the diurnal variation. Several reasons are mentioned in the Venkat Ratnam et al., (2014) and I suggest authors to go through the paper very carefully before making such statements.

**We agree that it was rather late to mention the previous work and they are now discussed at greater length in the introduction – after rereading Venkat Ratnam et al., (2014). The spelling is corrected.**

24. Lines 517-518. I do not agree with these statements. Note that descent winds are not from free floating of the balloon but dragged due to acceleration due to gravity in addition to the background wind.

**See next response.**

25. Lines 519-520. One cannot think of having accurate measurement of wind during descent as it is dragged by acceleration due to gravity that depends on left out busted balloon and weight of radiosonde in addition to the background wind.

**We disagree. The force of gravity is vertical so does not directly affect the horizontal motion of the radiosonde. During the ascent the radiosonde is pulled upwards by the buoyancy of the balloon (also gravity). If the radiosonde is falling fast then there will be a time lag before it reacts to a change in the horizontal wind – we mention this here using the shorthand 'inertial effects', see section 1 and the Appendix of Hock and Franklin.**

26. Line 521. Here references are needed for those studies. Also, I guess, those studies have been neglected due to known reasons (dragged winds rather than free floating along with background).

**We wrote "Most studies of radiosondes concentrate on the temperatures and humidities and the winds are somewhat neglected". To take one example, the section on winds in Dirksen et al (2014) is much shorter than the sections on temperature and humidity. Temperature and humidity are seen as higher priority for climate studies, they are more comparable with (most) satellite data, they are also more subject to biases than winds are (radar winds can have a direction bias). I think some authors see radiosonde winds as a 'solved problem', but pendulum motion deserves more attention. I think it would be overkill to give a long list of references at this point.**

27. Lines 525-527. Can one think of having a good quality balloon (almost sphere rather than oval after filling the gas) to make the balloon stable?

**I think the closest is the Jimsphere (which also has roughness elements), but I suspect that it wouldn't go as high in the stratosphere.**

28. Line 537. I doubt the descent winds for the reasons mentioned above.

**See response to point 26.**

29. Lines 539-540. I strongly suggest having a GRUAN descent product for the RS41 (for that matter any radiosonde type) as one more profile is being obtained at no cost. This is very useful particularly launched in the tropical altitudes as the expected diurnal variation is very high.

**We would like to see a GRUAN descent product, and this paper is a step in that direction. Resource and priority issues mean that it would take several years.**

30. Line 542. Same holds good even for T. Check the response time of the temperature sensor at higher descent speeds.

**See response on point 19.**

31. Line 546. I suggest providing a reference for this statement. Also, I do not agree that using parachutes will give some improvement.

**Figure 14 now referenced. Compare the German and Norwegian results (or UK and Finnish) to see the benefits of parachutes.**

32. I suggest to smooth the data with 100 m to take out the random motion of the balloon unless there is a need for very high vertical resolution. Further, 1s sampling particularly at higher altitudes (lower densities) may not be sufficient to sample the background atmosphere. I do not think that any NWP model requires parameters with very high vertical resolution for assimilation.

**In general, the data should be smoothed as little as possible leaving users to decide what processing best suits their application. NWP models do not need 1s sampling (but the vertical resolution of NWP models will improve in future). Some studies, including those of gravity waves, benefit from high vertical resolution (see Geller et al, 2021, and references).**

References:

Venkat Ratnam, M., N. Pravallika, S. Ravindra Babu, G. Basha, M. Pramitha, and B. V. Krishna Murthy (2014), Assessment of GPS radiosonde descent data, Atmos. Meas. Tech., 7, 1011–1025.

**Already referenced.**

---

## Author Comment (AC3)

**RC3** Review of manuscript by Ingleby et al., "On the quality of RS41 radiosonde descent data", submitted to Atmospheric Measurement Techniques

This paper presents interesting results describing the quality of RS41 radiosonde measurements made during descent after balloon burst, with and without a parachute. One finding that should be repeatedly stated in the manuscript is the reduction in positive temperature biases during descent when a parachute is used. An attempt is made to develop and apply a correction to these biases, based on descent rates, that includes descents with and without parachutes, but it seems like the very wide dynamic range of descent rates may require two correction equations - one for very fast descent rates (no parachute) and one for slower descent rates (parachute).

I had hoped for some deeper and more quantitative discussion of how bias-corrected RS41 descent data help improve NWP forecasts, but the paper is already quite long. I was also hoping for more results and discussion of ascent/descent RH comparisons, even though these are restricted to the troposphere and are therefore made more difficult by the high variability of RH below the tropopause.

**The impact on NWP forecasts seen was quite small – in part because Europe is already well-observed. Descent data from ships and remote islands should have more impact per profile, but would need quite long experiments to show a clear signal.**

**RH: we wondered if time in the stratosphere would adversely affect the RH sensor, but were pleasantly surprised to see that it doesn't. Apart from the Finnish descent RH results being slightly worse we didn't see any systematic differences (unlike winds and temperature). Yes, the variability of humidity in the troposphere is a complicating factor.**

In general, there are too many figures (27) in this manuscript, making it a lengthy and arduous read. Perhaps photographs of bursting (Fig 3) and burst (Fig 4) balloons can be moved to the supplement? I have also indicated in my comments where I think some figures can be eliminated.

**We know that the manuscript is long. We have moved Figure 4 to the supplement but included two images from it in a revised Figure 3 – we were reluctant to move all the photographs to the supplement. We have moved two other figures out of the main manuscript as suggested.**

My specific comments are:

Table 1: For me, numerical station codes are meaningless unless they are further identified by location in the caption. It would be interesting to know which stations launch the larger balloons, and why.

**Added "The names/locations of the stations can be found in https://oscar.wmo.int/oscar/vola/vola_legacy_report.txt ." (I wasn't sure if the reviewer wanted names or lat/long or both.)**

**Balloon size is a compromise between cost and the desire to get profiles as high as possible – different NMSs make different decisions. Some stations use automated launchers, the earlier systems could not cope with larger balloons, the latest systems can use most balloon sizes.**

Figure 7: This would be an excellent place to show the variability of descent rates in each country. A horizontal error bar (±1 std deviation) on each mean profile at 2-3 altitudes would be illustrative. Slightly offsetting the error bars in the vertical will allow them all to be viewed with clarity.

**I tried showing error bars for all countries but the plot became too 'busy'. As a compromise I have included the error bars for German stations (and changed the text accordingly).**

Line 183: When I calculate the reciprocal of 14.9 seconds I get 0.07 Hz (not 0.06 Hz).

**Changed to 0.07 Hz.**

Line 184: This sentence tends to imply that the increase (decrease) in period (±0.6 s) is linear with an increase (decrease) in tether length, which it is not.

**Sentence added to make the nonlinearity clear.**

Line 228: Here I think "ascent" should be changed to "descent"

**Changed to 'descent'.**

Line 239: The errors introduced into descent wind data are largely dependent on the vertical (temporal) averaging window. Errors can be made very small by using an averaging window that is similar to the period of the pendulum motion. If higher resolution wind data are required, the magnitudes of the errors become quite dependent on the averaging window.

**We agree, but using an optimal averaging window is non-trivial. We have added (in italics here) a note to the end of the previous paragraph: 'The amplitude of oscillations seen under this scenario could introduce error in the processed winds and is a possible area for future study *as is the optimal filtering for descent winds*.'**

Figure 11 and Line 283: To me, the "fit the background more closely" in line 283 implies that the mean O-B differences for descent in Figure 11 should be smaller than those for ascent, particularly at upper altitudes, which they are not. I think your statement in line 283 is instead referring to the variability of O-B differences, not the mean differences. "More closely" makes me think of reduced bias, not reduced variability. Perhaps "fit the background with less variability" is a better description of what is shown in Figure 11. Is less variability or smaller biases in O-B differences more important overall? I would think smaller biases.

**Changed to "fit the variations in background wind more closely". Both small bias and small standard deviation (SD) in O-B are important. The biases are small for both ascent and descent, the main difference is in the SDs.**

Line 289: What is the typical (range?) of averaging windows used to reduce pendulum effects on wind data? Do these change from flight to flight? For a given flight are the averaging windows different for ascent and descent?

**The averaging window does not change from flight to flight or ascent to descent. The text has been changed to 'In both cases a time filter with a fixed window is applied to all profile data. Improvements to this are possible as is clear from Section 2.5.'**

Figure 12 and Line 294: The low vertical resolution in Figure 12 makes it difficult to distinguish between pendulum motion and noise. Can you instead display some sections of the profiles at higher vertical resolution so the pendulum motion becomes more obvious?

**Two sections are now shown in more detail (it is now Figure 11).**

Line 297: Does "more pendulum motion" imply a higher frequency or a larger amplitude of oscillations? Which has the greater impact on the standard deviations of O-B averages?

**Changed to "larger amplitude pendulum motion". Which has the greater impact – probably the amplitude.**

Table 2: Are these statistics for all radiosonde sites, a subset of sites, or a single site? I assume "Sample" is the sample size for either RS-RO or RS-B observations, but don't know which. Adding the sample sizes for the observations not represented by "Sample" would be helpful (and will show your point about the smaller number of RS-RO differences.

**Added clarifications to the caption: "for all stations" and "the comparisons with the background are limited to the profiles collocated with RO".**

**Also added sample size to the mian text: "- note that the sample size is much smaller than for the O-B statistics (137 versus 2190 at 70 hPa)."**

Figures 14 and 15: I don't see the need for Figure 15 when Figure 14 clearly shows the warm temperature biases of the descent profiles. Lines 325-326 are adequate to explain why a simple time lag argument for the biases is not adequate.

**Figure 15 deleted.**

Line 331: "by 300 hPa" is confusing. Less confusing is "below 300 hPa".

**Changed to "below"**

Figures 18 and 19: I don't see the need to have a Figure each for specific and relative humidity differences when the radiosondes directly measure relative humidity. Also, specific humidity calculated from relative humidity measurements requires pressure and temperature data that can themselves produce biases in specific humidity values. It's difficult to argue that 2 separate figures are needed to support 4 sentences in the text.

**Figure 19 moved to supplement.**

Line 377: Why wasn't the Prague site included in Table 1 and Figure 7? Here, it just comes out of the blue as a site with many useful descent soundings.

**We have added Czechia to Table 1 and explained in the caption to Table 1 that the data were not provided to ECMWF in real-time (hence the absence from Figure 7). Some NMSs were/are reluctant to provide 'experimental data' via the GTS (especially using the dropsonde template).**

Line 378: What is a "comparable level"?

**Changed to "there were about 528 000 points at which ascent and descent could be compared"**

Line 381: Here, the change in writing style (and presumably authorship) is obvious. The lead author may want to reconstruct some of the sentences to flow as the previous text did and to help with clarity in this section.

**Yes, section 4 was written by MM. It had been rewritten in places to improve the flow/English, but more has now been done. Some of the figures have been modified to show mean differences as dashed lines and SDs as solid lines for consistency with section 3.**

Line 384: I have no idea what this means: "was expected warm bias for 06 and 12 terms, and cold bias for 00 term due to the diurnal variation.

**Much of this paragraph has been rewritten for clarity (shortening it where possible).**

Line 387: I presume "dividing the sample into groups of 1000 m" refers to the binning of differences by altitude using 1000 m-thick layers?

**Yes, rewritten as "After dividing the sample into bins of 1000 m in altitude".**

Line 390: Does "separating data into 00, 06 and 12 UTC groups" indicate that the Prague site launches 4x per day at 00, 06, 12 and 18 UTC? Why not simply say that" And say that the data from soundings performed at each of the four launch times were analyzed separately? If launches were performed at 18 UTC, why are those data not shown or discussed? I assume it is because of the large difference reduction in near-surface temperatures between 18 and 20 UTC, but this needs to be stated.

**There are three launches a day, this is now stated explicitly.**

Figure 21: Which launch time is this for?

**All launch times, now stated explicitly.**

Line 399: If not from a quadratic fit of the data in Figure 20, how was the "best estimate" determined? What is the uncertainty of A?

**The best estimate was determined as the minimum of the mean square difference of T between ascent as descent - so similarly as it would be from Figure 20, but for all 528000 original comparable points. Uncertainty of A wasn't determined (we were not sure how to do that for the quadratic coefficient).**

Line 401: "the root mean square ΔT is lowered from 1.22 °C to 1.06 °C, indicating that the correction explains 24.4 % of the variance seen". How does the reduction in RMS explain 24.4% of the variance? Isn't that determined from a correlation coefficient?

**Variance between ascent and descent temperatures is given by various parameters - three main seems to be descent rate implying friction, time and space difference of measurement and uncertainty of measurement itself. Measure of the variance is (mean) square difference. As the RMS of ΔT is lowered from 1.22 °C to 1.06 °C, MS of ΔT is lowered from 1,49 to 1,12, which is 24.4% decrease. The rest 1,12 belongs to the other parameters than friction.**

Table 3: Each coefficient needs an uncertainty estimate or their statistical significance remains unknown. "Best estimate" still needs to be explained since it appears to not be based on a standard parametric fit.

**See explanation above (reply to comment on line 399). I think that best estimate is quite standard.**

Table 4: What are "compared levels: 527 779" ?

**Changed to 'sample size'.**

Line 431: "the exact value of the correction coefficient is slightly uncertain." Are you suggesting that the correction should be the same for descents with and without a parachute? I don't understand how that is expected, since descent rates without a parachute are typically much higher than those

with a parachute. If you divide the data into types: high descent rates typical of no parachute descents and lower descent rates (parachute) and independently fit each type, can you improve the quadratic fits and lower the uncertainties in "A" for each case?

**There is also plenty of overlap in the descent rates with and without a parachute. We would prefer to use the same correction for both, unless there is strong evidence of the need for different corrections. Figure 19 (moved from section 3) supports the use of a single correction.**

Line 436: "method is used for processing of the data from RS41-SGP radiosondes" - what method are you referring to and how is it used to process the data? The equation is only used to calculate geopotential height.

**Reworded and simplified.**

Line 438: In order to calculate the pressure profile the geometric altitudes from GPS must be converted to geopotential heights. It is very important to mention this.

**Yes, changed to "The RS41-SG radiosonde measures geometric height using GPS, this is converted to geopotential height, and the pressure is calculated with the hydrostatic equation."**

Line 439: A radiosonde doesn't "estimate" anything, it measures descent temperatures with a positive bias.

**Reworded: "As discussed above, descent at high speeds, mostly in the stratosphere, causes the measured temperature to be too high." I have left the next sentence unchanged "This overestimation of temperature leads to underestimation of air density." as it seems to be the simplest/shortest way of describing the effect on the processing.**

Line 443: "the shift of height still remains in the troposphere levels" - why would you expect it to go away? The effect is cumulative during the entire descent so the geopotential height biases continue to increase down to the surface.

**Changed to "the shift of height affects the troposphere levels". This insight (from MM) is obvious once pointed out, but hadn't occurred to me earlier.**

Figure 23: It would be instructive to say that the vertical offsets between ascent and descent temperatures shown are for a RS41-SG and that the offsets are due to what is described in Lines 444-445.

**Now Figure 20. Text shortened and clarified.**

Lines 452-453: The descriptions of curve colors in Figure 24 belong in the caption, not here in the text. The caption for Figure 24 needs some rewording, including a statement of the Figure being based on a single flight or many flights.

**Colours removed from text and figure (now 21) replotted. Caption changed to clarify that it is based on many flights.**

Line 454: "and the lines are almost the same". Wouldn't it be clearer to say that the biases and standard deviations in differences between ascent and descent temperature measurements are the same regardless of whether the profiles are aligned using height or pressure?

**The suggested wording is much longer, we changed to: "In the stratosphere the choice of coordinate has little impact on the ΔT statistics (because ΔT comes mainly from direct heating)."**

Line 474: Would the "positive effect" be the reduction in biases between ascent and corrected descent measurements?

**This introductory sentence was deleted and the paragraph reworded in several places.**

Figure 26: The caption for Figure 26 needs some rewording.

**Reworded.**

Figure 27: The "% differences" units are misleading because they are not % differences. My initial thoughts were that the 100% line (control) was perfect agreement and deviations from this line represented positive and negative biases. Then the caption indicates that values <100% indicate improved forecasts. A clearer explanation is needed here to make the results shown in this figure more understandable.

**They are % differences. Text changed to 'modest improvements in the root-mean-square (rms) fit of the 12 h forecast to radiosonde ascent data (Figure 24; 100\*rms_test/rms_control is shown).' Rms is a natural measure for vector winds.**

Line 513: You might check with the US National Weather Service Field Support Laboratory in Sterling, VA, about publications arising from wind tunnel tests of radiosonde temperature sensors. They have decades of experience performing such tests in a wind tunnel environment.

**We have checked with someone at Sterling, and also someone at INRIM in Italy. Neither was aware of any such tests, but there was some interest in performing them.**

Line 519: "have a closer RMS fit" -  this is far from standard terminology. What is an "RMS fit"?

**Changed to "closer overall fit". (RMS - root-mean-square, quite often used in NWP, but simpler to avoid it here.)**

Line 528: "Jimsphere balloons" need to be described here

**Added 'One technique to obtain accurate wind profile data is the Jimsphere - a balloon with roughness elements used without a separate instrument package.' See https://spinoff.nasa.gov/spinoff1996/48.html. They were developed and used by NASA, but it is unclear to me how much they are still used.**

Line 529: If you have documented evidence that supports this statement please provide it here (along with the reference) otherwise this appears to be opinion rather than fact.

**Changed to "Dropsonde wind profiles suffer much less from pendulum motion than radiosonde ascent winds (Wang et al, 2008) and probably less than radiosonde descent winds."  This is discussed in a bit more detail in the introduction.**

Line 546: "this will reduce with improved processing/bias correction"  - it is not clear what "this" is.

**Changed to "the benefits of parachutes or pressure sensors will reduce with improved processing/bias correction".**

Lines 552-554: The text inside the parenthesis needs to be re-worked, as it is not clear what is meant.

**Changed to "(it would be better to exclude data based on the actual descent rate, but this would require more work)." Arguably data should be rejected for a while after the parachute has opened abruptly (see Fig 9, was Fig 10) but I think we'll leave this detail out.**

Line 557: Remove the space between "n" and "ow"

**No space in "now" - a quirk of the pdf?**